# Deciphering the Key Regulatory Roles of KLF6 and Bta-miR-148a on Milk Fat Metabolism in Bovine Mammary Epithelial Cells

**DOI:** 10.3390/genes13101828

**Published:** 2022-10-09

**Authors:** Ambreen Iqbal, Haibin Yu, Ping Jiang, Zhihui Zhao

**Affiliations:** The Key Laboratory of Animal Genetic Resource and Breeding Innovation, Department of Animal Sciences, College of Coastal Agricultural Sciences, Guangdong Ocean University, Zhanjiang 524088, China

**Keywords:** bovine mammary epithelial cells, bta-miR-148a, *KLF6*, milk fat

## Abstract

MicroRNAs (miRNAs) are non-coding RNAs that regulate the expression of their target genes involved in many cellular functions at the post-transcriptional level. Previously, bta-miR-148a showed significantly high expression in bovine mammary epithelial cells (BMECs) of Chinese Holstein cows producing high milk fat compared to those with low milk fat content. Here, we investigated the role of bta-miR-148a through targeting *Krüppel-like factor 6 (KLF6)* and further analyzed the role of *KLF6* in regulating fat metabolism through targeting *PPARA, AMPK/mTOR/PPARG,* and other fat marker genes in BMECs of Chinese Holstein. The bioinformatics analysis showed that the 3’ UTR of *KLF6* mRNA possesses the binding sites for bta-miR-148a, which was further verified through dual-luciferase reporter assay. The BMECs were transfected with bta-miR-148a-mimic, inhibitor, and shNC, and the expression of *KLF6* was found to be negatively regulated by bta-miR-148a. Moreover, the contents of triglyceride (TG), and cholesterol (CHO) in BMECs transfected with bta-miR-148a-mimic were significantly lower than the contents in BMECs transfected with bta-miR-148a-shNC. Meanwhile, the TG and CHO contents were significantly increased in BMECs transfected with bta-miR-148a-inhibitor than in BMECs transfected with bta-miR-148a-shNC. In addition, the TG and CHO contents were significantly decreased in BMECs upon the down-regulation of *KLF6* through transfection with pb7sk-*KLF6*-siRNA1 compared to the control group. Contrarily, when *KLF6* was overexpressed in BMECs through transfection with pBI-CMV3-*KLF6*, the TG and CHO contents were significantly increased compared to the control group. Whereas, the qPCR and Western blot evaluation of *PPARA, AMPK/mTOR/PPARG,* and other fat marker genes revealed that all of the genes were considerably down-regulated in the *KLF6*-KO-BMECs compared to the normal BMECs. Taking advantage of deploying new molecular markers and regulators for increasing the production of better-quality milk with tailored fat contents would be the hallmark in dairy sector. Hence, bta-miR-148a and *KLF6* are potential candidates for increased milk synthesis and the production of valuable milk components in dairy cattle through marker-assisted selection in molecular breeding. Furthermore, this study hints at the extrapolation of a myriad of functions of other KLF family members in milk fat synthesis.

## 1. Introduction:

Milk is a nutrient-rich white liquid obtained from mammals’ mammary glands, and it is a chief nutrient source for infant mammals [1]. Milk fat is a vital source of energy owing to the nutrients such as bioactive lipids and fat-soluble vitamins [2]. In milk fat, the principal component is triacylglycerol (TAG), which comprises about 97–98%. In contrast, the other minor components include diacylglycerol (DAG), which comprises approximately 0.28 to 0.59 percent of weight, and monoacylglycerol (MAG) has about 0.16 to 0.38 wt percent range. The free fatty acids are just about 0.1 to 0.44 wt %, while the CHO component is about 0.4 to 0.45%. [3].

MicroRNAs are one of the molecular players in regulating milk fat synthesis in mammary epithelial cells. MiRNAs belong to the family of non-coding RNAs that is about 22 nucleotides long. They regulate the expression of their target genes by binding to the 3’ untranslated region (3’ UTR) of their mRNAs [4]. The expression of miRNAs in the mammary gland arbitrates important functions, including mammary gland development, fat metabolism, and lactation [5,6]. Several researchers have investigated the roles of miRNAs in dairy lactation or mammary gland development. Recently, Do et al. found that 58 miRNAs were dynamically and differently expressed across lactation phases and that 19 miRNAs were significantly and time-dependently expressed during lactation [7]. Furthermore, lipid metabolism-related miRNAs’ role in lipid metabolism has been reported [8]. Several lipid metabolism-related miRNAs including miR-33 [9], miR-122 [10], miR-370 [11], miR-378/378 * [12], miR-143 [13], miR-335 [14], and miR-103 [15] have been reported. Moreover, miR-103 [15], miR-2885, miR-135a, and miRNA-370 are involved in the metabolism of glucose and lipids [16].

The miR-148a belongs to a highly conserved family, the miR-148/miR-152 family, and is abundantly expressed in mammary gland tissue [17,18]. Previous studies reported the role of miR-148a in the differentiation of the C2C12 myoblast and skeletal muscle cells and myoblast differentiation into the myotube [19]. Van Wijnen et al. [20] stated that miR-148a was implicated in osteoclast development and caused the monocyte to osteoclast transition. Gailhouste et al. [21] indicated that improved miR-148a expression could induce liver cell differentiation and maturation by the inhibition of DNA (cytosine-5-)-methyltransferase 1 (*DNMT1*). The high expression of miR-148a in the adult liver controls cholesterol and triglyceride homeostasis [22]. In addition, miR-148a could facilitate the differentiation of primary adipocytes to mature adipocytes [23]. Additionally, adipocyte differentiation and regeneration are both affected by miR-148a and miR-17-5p [24]. Remarkably, miR-148a and miR-17-5p seemed to increase the triglyceride content and lipid accumulation of goat MECs [25]. In addition, miR-148a also shows high milk expression as a component of milk exosomes and milk fat globules [26]. Intriguingly, single-nucleotide polymorphisms (SNPs) in the promoter region of miR-148a have recently been linked to changing LDL-C and triglyceride levels in humans, implying a putative physiological role for this miRNA in regulating lipid metabolism and thus emphasizing it for further research [27,28,29]. Bta-miR-148a also showed significantly high expression in BMECs from high-fat Chinese Holstein BMECs compared to the low-fat Chinese Holstein BMECs [30].

Krüppel-like factors (KLFs), a family of transcription factors (TFs), bind preferably to the GC-rich sequence CACCC through its three conserved zinc finger motifs toward the C-terminal side of 23–25 amino acids [31]. KLF6 binds to different acetyl-transferases, including cyclic adenosine monophosphate response element-binding protein (*CREBBP*), p300, and p300/CBP-associated factor [32]. The binding of KLF6 to these acetyl-transferases leads to the acetylation of *KLF6*, which stimulates its transcriptional activity [33]. It can also lead to the acetylation of histones turning into chromatin remodeling and transcription initiation in areas targeted by KLF6 [33]. Furthermore, the nuclear receptor peroxisome proliferator-activated receptor α (*PPARA*) regulates the expression of genes related to FA oxidation, lipid uptake, lipid transport, and β-oxidation [34]. Through their engagement with *PPARA*, three members of the KLF family enhance FA oxidation in the heart [34].

The *PPARs* belong to the nuclear receptor family and ligand-inducible transcription factors [35]. The family of *PPARs* is made up of three members: *PPARA* (also known as NR1C1), *PPARB* (also known as NR1C2), and *PPARG* (also known as NR1C3). *PPARs* are combined with retinoid X receptors to form heterodimers. They govern the network gene expression engaged in several processes such as lipid metabolism, adipogenesis, metabolic balance, and inflammation [36]. The first *PPAR* to be recognized as *PPARA* is located in the liver and heart. Hypolipidemic fibrate medications decrease cholesterol levels by targeting this enzyme, which regulates fatty acid oxidation [37]. *PPARs* are also present in brown adipose kind of fat. Note that whereas *PPARG* (also known as *PPARB* and more generally referred to by its abbreviation *PPARG/B*) shares many of the same activities as its counterpart, it is articulated in a wider range of tissues, including the heart, skeletal muscle, and liver [36,37]. *PPARG* is a master controller of adipogenesis and an effective modulator of whole-body insulin sensitivity and lipid metabolism [35,38]. Due to an alternative promoter and splicing use, *PPARG* has two isoforms, *PPARG1*, and *PPARG2*, at its N terminus having an extra 30 amino acids [35,38,39]. High fat diet can stimulate the expression of *PPARG2* in various organs, although *PPARG*1 is expressed in numerous tissues [39,40].

In vitro cultured bovine mammary epithelial cells (BMECs) are widely used as an experimental cell model for studying milk production or its components due to their milk synthesizing ability. Previously, our group reported the role of different miRNAs in regulating the expression of milk-related genes in BMECs. Therefore, BMECs were used in this study as the experiment object to verify the relationship between bta-miR-148a and the target gene *KLF6*. The role of bta-miR-148a on the content of triglyceride, cholesterol, and free fatty acids in BMECs after the overexpression or inhibition of bta-miR-148a and by the construction of pBI-CMV3-*KLF6*, pBI-GFP-Neo-CMV3, pb7sk-*KLF6*-siRNA, and pb7sk-GFP-Neo was also investigated. To the best of our knowledge, this is the second study to report the role of bta-miR-148a in regulating fat metabolism. In addition, it is the first study in BEMCs that elucidates the role of *KLF6* in regulating fat metabolism through targeting *PPARA, AMPK/mTOR/PPARG,* and other fat marker genes in BMECs of Chinese Holstein.

## 2. Materials and Methods

### 2.1. Cell Source

The BMECs of Chinese Holstein dairy cows were isolated and maintained in the animal genetics and breeding laboratory, college of Animal Science, Jilin University [41] by following the guidelines for the care and use of laboratory animals of Jilin University (Animal Care and Use Committee permit number: SY201901007) [42,43,44].

### 2.2. Experimental Plasmids and Reagents

The plasmids used in this study include bta-miR-148a mimics, inhibitors, and negative controls synthesized by (Shanghai Gene Pharma Corporation Technology Co., Ltd., Shanghai, China) and cloned in *E. coli* DH5α cells (Vazyme, C502-03 Nanjing, China). The plasmid was extracted using (Endo free Maxi Plasmid kit, Tiangen Biotech Co., Ltd., DP117, Beijing, China). Transfection was performed using Fugene HD Transfection Reagent (Promega Biotechnology Co., Ltd., Beijing), Total RNA Extraction Reagent TRIzol (15596-026, Invitrogen, Waltham, MA, USA), a Reverse Transcription Kit (Vazyme, Hiscript II qRT SuperMix for qPCR (gDNA wiper) R223-01 China), and qPCR (Vazyme, ChamQ Universal SYBR qPCR Master Mix, Q711, Nanjing, China). Primers were synthesized by Genewiz (Suzhou, China). Chloroform, absolute ethanol, and isopropanol of the domestic analytical pure category were from Tianjin Concord Technology. Xho I restriction enzyme (K2704AA) was purchased from TaKaRa, Not I restriction enzyme (0551712) was purchased from Biolabs, and T4 DNA ligase (00758655) was purchased from Thermo Scientific. PBS was obtained from Sangon Biotech (GC10FA0002, Shanghai, China). Protein was extracted using RIPA lysis buffer (Dalian Meilun Biological Technology Co. Ltd., Bejing, China, PN: MA0171). Protein concentration was determined with a TaKaRa BCA Protein Assay Kit, (T9300A) and *KLF6* antibody (bs-1395R, Bioss Co. Ltd.). A dual-luciferase reporter assay kit (Vazyme, cat# DL101-01, Nanjing, China), triglyceride assay kit (Applygen, Beijing, China), and cholesterol assay kit (Applygen, Beijing, China) was used during the whole research work.

### 2.3. Experimental Equipment

The following experimental apparatus and instruments were used in this study: 5% CO_2_ cell culture incubator (Thermo, Marietta, OH, USA) and a fluorescent microscope (TE2000, Nikon, Tokyo, Japan). cDNA was prepared on a Bio-Rad T100 thermal cycler, qPCR was performed using the Bio-Rad CFX connect real-time system (USA), and the quality of RNA was measured using a spectrophotometer (UNIC2802H, Shanghai, China). The Wet/Tank Blotting Systems (Bio-rad, Hercules, CA, USA) were used to perform the Western blotting. The signal intensities were captured by a Tanon 5200 chemiluminescence/fluorescence image analysis system (Tanon, Urumqi, China). The luciferase activity and TG contents were detected by the SpectraMax M5 microplate reader (Molecular Devices, San Jose, CA, USA). Other apparatus involved in this study included a liquid nitrogen tank (Chart/Golden Phoenix Liquid Nitrogen Container USA), Microplate Centrifuge (Tiangen Biotech Co., Ltd., Beijing, China), and Gel Imager (Alpha Innotech, San Leandro, CA, USA).

### 2.4. Bioinformatics Prediction of bta-miR-148a Target Gene

The mature sequence of bta-miR-148a was identified from (http://www.mirbase.org (accessed on 6 February 2021)), and the binding sites for bta-miR-148a in 3’ UTR of *KLF6* were found through TargetScan (http://www.targetscan.org (accessed on 10 February 2021)). The primers for qPCR (Table 1) and luciferase activity (Table 2) used in this study were designed by Primer Premier 6.0 (Premier Biosoft International, Canada). Each experimental step/reaction was repeated three times, and *GAPDH* or *U6* were used as a reference where needed. Primers (Table 1) were designed for the 3’ UTR, and PCR amplified the 3’ UTR of *KLF6*.

### 2.5. Luciferase Reporter Assay

The bioinformatic analysis predicted that *KLF6* has two bta-miR-148a binding sites in its 3’ UTR. PCR amplified the 3’ UTR sequence of *KLF6* for Not I and Xho I restriction fragments. Then, the amplified sequence was cloned into a pmiR-RB-REPORT^TM^ vector to construct the vector of pmiR-RB-REPORT^TM^-*KLF6*-WT/MUT. The sequences for the DNA oligos used in this particular assay are listed in Table 2. These vectors of pmiR-BR-REPORT^TM^-*KLF6*-WT/MUT were co-transfected with bta-miR-148a-mimic into the BMECs using the same transfection conditions mentioned above except that the used concentration of pmiR-BR-REPORT^TM^-*KLF6*-WT/MUT and bta-miR-148a-mimic was 1 μg. The Dual-Glo luciferase assay system (Promega, USA) was used to detect the firefly (hluc+) and Renilla (hRluc) luciferase activities according to the manufacturer’s protocols. The hluc+ luciferase was used as the reference to correct the transfection efficiency variation, and the relative luciferase activities were calculated as hRluc/hluc+. A spectroMax M5 microplate reader detected the fluorescence values, and the results were analyzed using GraphPrism8 software. This experiment was conducted on three replicates with the same number of cells and transfected with the same vectors using the same culture conditions.

The *KLF6* overexpression vector, pBI-CMV3-*KLF6*, was a kind gift from Lixin Xia of Jilin University. For the *KLF6* interference vector construction, two complementary oligos with restriction cutting sites on the 5′ end was designed using the siRNA Construct Builder tool of GenScript Biotech and synthesized by Sangon Biotech (Shanghai; Table 3). For the siRNA cloning, 10 µL of the reaction mixture was made from 4 µL of NEB buffer, 4 µL of ddH_2_O, 1 µL of F, and 1 µL of R primer. Then, this reaction mixture was put in the T100 Thermal Cycler of Bio-Rad for the annealing purpose using different temperature gradients for specific times including 90 ℃ for 5 min, and 80 ℃, 70 ℃, 60 ℃, 50 ℃, 40 ℃, 30℃, 20 ℃, and 16 ℃ all for 30 s each. For the ligation purpose, 10 µL of the reaction mixture was prepared from 1 µL of T4 ligase buffer, 2 µL of annealed product from the previous step, 5 µL of pb7sk-GFP-Neo, and 2 µL of ligase buffer. We kept this reaction mixture for 5 h at 16 ℃. After 5 h, 50 µL of DH5α cells were taken, and 10 µL of connecting liquid was added and placed in ice for 30 min after gentle mixing. Then, the mixture was placed in the water bath at 42 ℃ for 60–90 s and placed back on the ice for 3 min. Following this step, 450 µL of LB media was added, and the mixture was put for 45 min at 37 ℃. The mixture was then spread uniformly on the solidified LB media in a Petri plate. The Petri plate was placed in an incubator at 37 ℃ for 12–16 h. Following the growth of bacterial colonies, the good colonies were selected and put in LB medium, which was shaken for 5 h at 37 °C. About 500 µL of sample from turbid solution was sent to Sangon Biotech for sequence matching. After sequence matching, the plasmid was extracted using the EndoFree Maxi Plasmid kit (Tiangen Biotech Co., Ltd. Beijing, China) following the supplier’s instructions.

### 2.6. Cell Culture and Transfection Reagents

The culture conditions for BMECs included basal media consisting of DMEM/F12 (Hyclone, Grand Island, NY, USA) supplemented with 10% (*v*/*v*) fetal bovine serum (Pasching, Austria), and the cells were incubated with 5% CO_2_ at 37℃. For the transfection of bta miR-148a-mimic, inhibitor, and shNC plasmids, the BMECs were seeded in a six-well cell culture plate (Nest, Wuxi, China) with about 1 × 10^6^ cells/well. The old culture media was removed upon reaching 70–80% cell confluency, BMECs were washed twice with PBS (Sangon Biotech, Shanghai, China), and fresh culture media was added. The transfection system included 2 μg of plasmid, 6 μL of Fugene, and 200 μL of DMEM/F12 for each well. After mixing, the transfection mixture was incubated for 15 min at room temperature, then transferred to the BMECS in a six-well plate, and cells were put back in the incubator. After 24 h, the cells were examined under a fluorescence microscope for cell morphology and evaluation of transfection through the expression of green fluorescent protein (GFP). There was a total of three replicates with the same number of cells and the same culture and transfection conditions. The same protocol was used for the transfection of pb7sk-*KLF6*-siRNA, pb7sk-GFP-Neo, pBI-CMV3-*KLF6*, and pBI-GFP-Neo-CMV3 into BMECs.

### 2.7. Extraction of RNA and qPCR

After 24 h of transfection of BMECs with miR-148a-mimic, miR-148a-inhibitor, and miR-148a-negative control (shNC) vectors, the total RNA was extracted using the TRIzol reagent following the manufacturer’s protocol. The purity and RNA concentration was determined by agarose gel electrophoresis and spectrophotometer (Thermo NANODROP-2000). Then, 1 μg of total RNA was reverse transcribed to cDNA using the Reverse Transcription Kit (VAZYME, Hiscript II qRT SuperMix for qPCR (gDNA wiper).

For the *KLF6* overexpression, interference, and control group of BMECs after 24 h of respective transfection with pBI-CMV3-*KLF6*, pBI-GFP-Neo-CMV3, pb7sk-*KLF6*-siRNA, and pb7sk-GFP-Neo, the same methods were used for RNA extractions, purity and concentration measurement, and cDNA synthesis was used in the bta-miR-148a experiment.

Real-time quantitative PCR was performed in a 20 μL reaction mixture including the following components: 10 μL of SYBR Mix, 0.5 μL of each of the upstream and downstream primers, 1 μL of cDNA, and 8 μL of ddH_2_O. The reaction procedure included the following step; pre-denaturation at 95 ℃ for 30 s, denaturation at 95 ℃ for 5 s, and annealing at 60 ℃ for 30 s with 35 cycles of repetition. The final values were calculated through 2^−ΔΔCt^ by using *GAPDH* as a reference. GraphPadPrism8 software was used for the statistical analysis, and the unpaired *t*-test function was used to compare different groups for differential expression analysis. All experiments were conducted in three replicates using the same number of cells and culture and transfection conditions. Real-time PCR primer sequences of PPARG and PPARA pathway related, and the marker genes of lipid synthesis are shown in Table 4. 

### 2.8. Screening of KLF6 Gene Interference Vector

Total RNA from BMECs was extracted 24 h after transfection through the steps mentioned above RNA extraction. After that, the total RNA concentration was detected, and cDNA was reverse transcribed using the Reverse Transcription Kit (VAZYME, Hiscript II QRT SuperMix for qPCR (gDNA wiper). Using cDNA as a template, the SYBR Premix Ex-Taq ^TM^ fluorescence quantitative kit was used for qRT-PCR.

### 2.9. Western Blot Analysis

The total protein was extracted from BMECs 24 h post-transfection of bta-miR-148a-mimic, bta-miR-148a-inhibitor, and bta-miR-148a-shNC. Protein was extracted using RIPA lysis buffer (Dalian Meilun biological technology Co. Ltd., PN: MA0171). First, the BMECs were washed twice with PBS, and 200 μL of RIPA/well of a six-well cell culture plate was added, and cells were incubated at 4 ℃ for 30 min. Then, the cell mixture was collected in 1.5 mL centrifuge tubes and centrifuged at 12,000 rpm for 20 min. The supernatant was collected, and protein concentration was determined with a TaKaRa BCA Protein Assay Kit, T9300A). Then, Western blotting was performed by making SDS-PAGE (Epizyme, Biotechnology Co., Ltd. Shanghai, China, PG112) gel electrophoresis, and protein was run on the gel. For immunoblotting, primary polyclonal-KLF6 antibody and monoclonal GAPDH at a 1:500 dilution and a 1:1000 dilution were used, respectively. The secondary antibody used was the HRP-conjugated anti-rabbit antibody with 1:2000 dilution (BioWorld, Irving, TX, USA, BS13271). The signal intensities were captured by a Tanon 5200 chemiluminescence/fluorescence image analysis system (Tanon, China). This experiment was performed on three replicates using the same number of cells and culture and transfection conditions. The same protocol was used for the Western blot of KLF6 protein in pb7sk-*KLF6*-siRNA1, pb7sk-GFP-Neo, pBI-CMV3-*KLF6*, and pBI-GFP-Neo-CMV3 transfected BMECs.

### 2.10. Detection of the Intracellular Triglyceride Content

The BMECs were transfected with bta-miR-148a-mimic, inhibitor, and shNC in six-well cell culture plates to determine TG contents. After 24 h of successful transfection, the intracellular triglyceride content of BMECs was extracted and detected with the help of a tissue and cell triglyceride assay kit (APPLYGEN, E1013, Beijing, China) following the manufacturer’s protocols. The extracted samples were examined using the software Gen5 CHS (SYNERGY|HTX multi-mode reader, Bio Tek, S1LFTA), and the total TG content was adjusted by the quantity of total protein. Each sample was replicated thrice, and the final results were calculated using the average values. Moreover, this experiment was repeated three times following the same steps mentioned here. The same protocol was used for the intracellular triglyceride content of BMECs transfected with pb7sk-*KLF6*-siRNA1, pb7sk-GFP-Neo, pBI-CMV3-*KLF6*, and pBI-GFP-Neo-CMV3.

### 2.11. Detection of the Intracellular Cholesterol Content

For the determination of CHO, the BMECS were transfected with bta-miR-148a-mimic, inhibitor, and shNC in six-well cell culture plates. The intracellular cholesterol content of BMECs was extracted and detected by the tissue and cell cholesterol assay kit (APPLYGEN, China, E1015) following the manufacturer’s protocols. The extracted samples were examined using the software Gen5 CHS (SYNERGY|HTX multi-mode reader, Bio Tek, S1LFTA), and the total CHO content was adjusted by the quantity of total protein. Each sample was replicated thrice, and the final results were calculated using the average values. Moreover, this experiment was repeated three times following the same steps mentioned here. The same protocol was used to determine CHO in pb7sk-*KLF6*-siRNA1, pb7sk-GFP-Neo, pBI-CMV3-*KLF6*, and pBI-GFP-Neo-CMV3 transfected BMECs.

### 2.12. Statistics and Analysis

All results were presented as the means ± standard error of the mean (SEM) of separate experiments (*n* ≥ 3). The significant differential analysis among different groups was performed using unpaired *t*-tests in GraphPad Prism8 software (San Diego, CA, USA). Statistical significance is presented as * *p* <0.05, ** *p* <0.01, *** *p* <0.001.

## 3. Results

### 3.1. Expression Trend of bta-miR-148a in BMECs

After 24 h of transfection of BMECs with bta-miR-148a mimic, inhibitor and shNC, the BMECs were observed under a fluorescence microscope to observe the cell morphology and transfection efficiency. The morphology of cells remained unchanged, and the obvious expression of GFP verified the successful transfection (Figure 1A–C) and the transfection efficiency of 80–90% in the transfected group was observed. The successful transfection witnessed through GFP was further verified by investigating the relative expression of bta-miR-148a in BMECs transfected with bta-miR-148a mimic bta-miR-148a inhibitor and the shNC group using qPCR. The results about the expression trend of bta-miR-148a suggest that its relative expression in bta-miR-148a-mimic transfected BMECs was significantly higher (*p <* 0.001) in comparison to that of the shNC group with a 1.4-fold change between bta-miR-148a-mimic and the shNC group (Figure 2). On the other hand, a significantly lower (*p <* 0.001) relative expression of bta-miR-148a-inhibitor transfected BMECs as compared to that of the shNC group was observed at a 0.6-fold change (Figure 2). These results manifest that the plasmids for bta-miR-148a-mimic, inhibitor and shNC were successfully transfected and showed their respective expression in the BMECs.

### 3.2. Bta-miR-148a Regulates the mRNA Expression of KLF6

Over a dozen target genes linked to lipid metabolism were screened in addition to *KLF6* via bioinformatics prediction. From these target genes predicted in silico, we selected *KLF6* as a candidate target gene for finding its role in milk fat metabolism in BMECs of Chinese Holstein cattle. The relative mRNA expression of *KLF6* showed that compared with the bta-miR-148a-shNC group, bta-miR-148a significantly down-regulated (*p <* 0.001) the *KLF6* expression in bta-miR-148a-mimic transfected BMECs with 4-fold change (Figure 3). Meanwhile, the relative expression of *KLF6* was significantly up-regulated (*p <* 0.001) in BMECs transfected with bta-miR-148a-inhibitor as compared to that of the bta-miR-148a-shNC transfected BMECs with 4-fold change (Figure 3).

### 3.3. Bta-miR-148a Targets the KLF6 mRNA by Specifically Binding to Its 3’UTR

In the 3’UTR of *KLF6* gene mRNA, the bta-miR-148a binding site predicted in silico (Figure 4A, B) was further validated by the renilla luciferase assay. For this purpose, the pmiR-RB-REPORT^TM^-*KLF6*-3’-UTR-WT and pmiR-RB-REPORT^TM^-*KLF6*-3’-UTR-MUT were, respectively, co-transfected with bta-miR-148a-mimic into BMECs. The results showed that the relative luciferase activity in the *KLF6*-WT+bta-miR-148a-mimic group was significantly decreased (*p <* 0.001) compared to the *KLF6*-MUT+bta-miR-148a-mimic group. On the other hand, there was a significant increase (*p <* 0.001) in luciferase activity in the *KLF6*-MUT+bta-miR-148a-mimic group compared to the *KLF6*-WT+bta-miR-148a-mimic group with 4-fold change (*p <* 0.001; Figure 4C). The combined analysis of bioinformatic data and luciferase activity shows that bta-miR-148a directly targets the 3’-UTR sequence of *KLF6* mRNA with off-target activity identified in the mutant 3’-UTR sequence.

### 3.4. miR-148a Regulates the KLF6 Protein Expression

Western blotting was carried out to check the effects of transfection of bta-miR-148a-mimic, inhibitor, and shNC on protein expression of KLF6. The findings reflected that the pattern of KLF6 protein expression in BMECs transfected with the bta-miR-148a-inhibitor was higher relative to the mimic and shNC group at a 2-fold change (Figure 5). Meanwhile, the KLF6 protein expression was lower in the bta-miR-148a-mimic group than in the inhibitor and shNC group at 2-fold change. In addition to the mRNA expression of *KLF6* and luciferase reporter assay, these findings further reveal that *KLF6* is negatively regulated by bta-miR 148a.

### 3.5. Effect of bta-miR-148a on Triglyceride and Cholesterol Contents in BMECs

The synthesis of triglycerides in BMECs of Chinese Holstein cow specifically influences milk fat composition. TG content analysis of transfected BMECs revealed a substantial (*p* < 0.05) decrease in the TG content of BMECs in the bta-miR-148a-mimic group relative to the bta-miR-148a shNC group. Contrarily, TG contents in BMECs transfected by the bta-miR-148-a-inhibitor were substantially higher (*p* < 0.01) than that of the bta-miR-148a-shNC transfect BMECs (Figure 6A). The analysis of cholesterol content in transfected BMECs showed that the cholesterol content in BMECs of the bta-miR-148a-mimic group was low compared to the bta-miR-148a shNC group. However, significantly high levels of cholesterol expression were found in bta-miR-148a-inhibitor transfected BMECs (*p* < 0.01) compared to bta-miR-148a-shNC transfected BMECs with a 4-fold change (Figure 6B). These findings suggest that bta-miR-148a negatively regulates TG and CHO contents in BMECs.

### 3.6. The Relative mRNA Expression of KLF6 in Overexpression and Interference Groups

After the vectors of the pBI-CMV3-*KLF6* gene were transfected into the cells, the total RNA of the cells was extracted for reverse transcription and Real-Time Quantitative Reverse Transcription PCR (qRT-PCR). As shown in Figure 7A, compared with the control group pBI-GFP-Neo-CMV3, the relative mRNA expression of *KLF6* in pBI-CMV3-*KLF6* was significantly higher (*p <* 0.001). For the *KLF6* down-regulation study, three interference vectors of the *KLF6* gene were transfected into the cells, respectively, and the total RNA of the cells was extracted for reverse transcription and qRT-PCR. The results showed that upon a comparison with the pb7SK-GFP-Neo, the expression of *KLF6* in pb7SK-*KLF6*-siRNA1 was significantly lower (*p <* 0.01) with a 4-fold change, showing the best interference efficiency. Meanwhile, the relative mRNA expression in pb7SK-*KLF6*-siRNA2 and pb7SK-*KLF6*-siRNA3 was not significantly lower than the pb7SK-GFP-Neo. Therefore, the interference vector pb7SK-*KLF6*-siRNA1 was used for subsequent experiments (as shown in Figure 7B).

### 3.7. Cell Morphology and Transfection Efficiency of Overexpression and Inference Vectors

At 80% confluency, the BMECs were transfected with pBI-CMV3-*KLF6* and pb7SK-*KLF6*-siRNA1 vectors, separately. After 24 h, the cell morphology and GFP expression were observed by fluorescence microscopy. The morphology of cells remained the same after transfection, and the transfection of pBI-CMV3-*KLF6*, pBI-GFP-Neo-CMV3, pb7SK-*KLF6*-siRNA1, and pb7SK-GFP-Neo was successful with over 60-70% transfection efficiency, which was ample to be used for subsequent experiments (Figure 8).

### 3.8. Effect of pBI-CMV3-KLF6 and pb7sk-KLF6-siRNA1 on Triglyceride Contents in BMECs

TG contents in BMECs transfected by pBI-CMV3-*KLF6* were substantially higher (*p <* 0.001) than that of the pBI-GFP-Neo-CMV3 transfected BMECs (Figure 9A). These findings suggest that the overexpression of *KLF6* positively regulates TG contents in BMECs. Meanwhile, TG contents in BMECs transfected by pb7SK-*KLF6*-siRNA1 were substantially lower (*p <* 0.01) than that of the pb7SK-GFP-Neo transfected BMECs (Figure 9B), suggesting that the down-regulation of *KLF6* through pb7SK-*KLF6*-siRNA1 negatively regulates TG contents in BMECs. The TG contents in BMECs transfected by pBI-CMV3-*KLF6* were substantially higher (*p <* 0.01) than those of the pb7SK-*KLF6*-siRNA1 as a 4-fold change (Figure 9C). These findings suggest that the overexpression of *KLF6* increased the TG content compared to pb7SK-*KLF6*-siRNA1.

### 3.9. Effect of KLF6 Overexpression and Down-Regulation on Cholesterol Contents in BMECs

The analysis of cholesterol content in BMECs showed that the cholesterol content in pBI-CMV3-*KLF6* transfected BMECs was significantly higher (*p <* 0.01) than that in the pBI-GFP-Neo-CMV3 transfected BMECs (Figure 10A). This evidence indicates that pBI-CMV3-*KLF6* positively regulates the cholesterol contents in BMECs. The analysis of cholesterol content in transfected BMECs showed that the cholesterol content in BMECs transfected with pb7SK-*KLF6*-siRNA1 was significantly lower in comparison to the levels of cholesterol found in pb7SK-GFP-Neo BMECs (*p <* 0.001) compared to transfected BMECs (Figure 10B). This evidence indicates that pb7SK-*KLF6*-siRNA1 negatively regulates the cholesterol contents in BMECs. The cholesterol contents in BMECs transfected by pBI-CMV3-*KLF6* were substantially higher (*p <* 0.01) than those of the pb7SK-*KLF6*-siRNA1 as a 4-fold change (Figure 10C). These findings suggest that the overexpression of *KLF6* increased the CHO content compared to the pb7SK-*KLF6*-siRNA1.

### 3.10. Expression of KLF6 Protein in BMECs with KLF6 Overexpression and Down-Regulation

Western blotting was used to assess KLF6 protein overexpression and down-regulation in BMECs. The findings showed that the pattern of *KLF6* protein expression in BMECs transfected with the pBI-KLF6-CMV3 was higher relative to the pBI-GFP-Neo-CMV3 group (Figure 11A). The effects of transfection of pb7sk-KLF6-siRNA1 and pb7sk-GFP-Neo on the protein expression of KLF6 reflected that the pattern of KLF6 protein expression in BMECs transfected with the pb7sk-GFP-Neo was higher relative to the pB7sk-KLF6-siRNA1 group as a 2-fold change (Figure 11B). A similar trend of *KLF6* expression in these groups was seen at the mRNA level, and these results further verified this trend of KLF6 expression at the protein level (Figure 11B).

### 3.11. PPARG Relative mRNA Expression in Knock-out KLF6 and Normal BMEC Cell Line

The KEGG pathway (https://www.genome.jp/pathway/bta04152 (accessed on 1 January 2022) reported that *AMPK* negatively regulates the *mTOR* associated with the *PPARG* pathway (Figure 12A). The BMECs were cultured in 6-well plates. After 24 h, when the cells reached 90–95% growth, the total RNA was extracted and reverse transcribed, and qRT-PCR was performed. The relative mRNA expression of *KLF6-*KO-BMECs suggested that the expression of the *AMPK* was notably up-regulated (*p <* 0.001) in *KLF6*-KO-BMECs compared with the normal BMECs. Meanwhile, the mRNA expression of the *mTOR* was down-regulated in *KLF6*-KO-BMECs compared with the normal BMECs (Figure 12B), and the mRNA expression of the *PPARG* suggested that the expression of the *PPARG* in the *KLF6-*KO-BMECs was significantly down-regulated (*p <* 0.001) compared to the normal BMECs with the 4-fold change between the two groups. The mRNA expression results strongly suggested that *KLF6* through *AMPK/mTOR* targets the *PPARG* pathway and plays a milestone in controlling lipid synthesis in BMECs. These results also have the same trend, which elucidates the KEGG pathway.

### 3.12. Relative Protein Expression of the PPARG Pathway Genes in Knock-Out KLF6 and Normal BMEC Cell Line

The BMECs were cultured in six-well plates. After 24 h, when the cells reached 90–95% growth, the protein was extracted, and a Western blot was performed. The relative protein expression of *KLF6*-KO-BMECs demonstrated that the AMPK was highly up-regulated (*p* < 0.01) in *KLF6*-KO-BMECs than in the normal BMECs. Meanwhile, the protein expression of the mTOR was down-regulated (*p <* 0.01) in *KLF6*-KO-BMECs compared with the normal BMECs (Figure 13A). In contrast, PPARG protein expression revealed that PPARG expression was down-regulated in *KLF6*-KO-BMECs relative to normal BMECs. The protein results strongly suggested that KLF6 through AMPK/mTOR targets the PPARG pathway and plays a milestone in controlling fat synthesis in BMECs. These results also have the same trend, which elucidates the KEGG pathway.

### 3.13. Relative mRNA Expression of PPARA Pathway-Related Gene in Knock Out KLF6 and Normal Cell Line

The KEGG pathway (https://www.genome.jp/pathway/bta03320 (accessed on 1 January 2022)) reported the PPARA pathway (Figure 14A). The BMECs were cultured in six-well plates. After 24 h, when the cells obtained 90–95% growth, the RNA was extracted and reverse transcribed, and qRT-PCR was performed. The relative mRNA expression of *KLF6*-KO-BMECs showed that the *PPARA* was significantly down-regulated (*p* < 0.001) in *KLF6*-KO-BMECs compared with the normal BMECs. So, the *PPARA* pathway genes, which are involved in lipogenesis and cholesterol metabolism, were also investigated, finding that the relative mRNA expression of the *SCD* was down-regulated (*p <* 0.001) in *KLF6*-KO-BMECs compared with the normal BMECs. Meanwhile, the mRNA expression of the *MEI*, *CYPA1, CYP27A1,* and *LRX* suggested that the expression of the *MEI, CYPA1, CYP27A1,* and *LRX* in the *KLF6*-KO-BMECs was significantly down-regulated compared to the normal BMECs with 4-fold change. The expression of the *PPARA* pathway genes was significantly down-regulated in the *KLF6*-KO-BMECs, which validates the importance of the *KLF6* gene. The mRNA results strongly suggested that *KLF6* targets the *PPARA* pathway, and *PPARA* has a milestone role in regulating fat synthesis in BMECs. These results elucidate that the *KLF6* through the *PPARA* pathway regulates lipid synthesis in BMECs, as shown in Figure 14B.

### 3.14. Protein Expression of the PPARA in Knock-Out KLF6 and Normal BMEC Cell Line

The BMECs were cultured in six-well plates. After 24 h, when the cells attained 90–95% growth, the total protein was extracted, and the Western blot was performed. The protein expression of *KLF6*-KO-BMECs showed that the PPARA was significantly down-regulated (*p* < 0.001) in *KLF6*-KO-BMECs compared with the normal BMECs. Meanwhile, the protein expression of the SCD, MEI, CYPA1, CYP27A1, and LRX was significantly down-regulated in *KLF6*-KO-BMECs compared with the normal BMECs. The Western blot results strongly suggested that KLF6 targets the PPARA pathway and has a milestone role in controlling lipid production in BMECs. These results elucidate that KLF6 through the PPARA pathway regulates the lipid synthesis in BMECs, as shown in Figure 15A.

### 3.15. The mRNA Expression of Marker Genes of Lipid Synthesis in KLF6-KO-BMECs

The BMECs were cultured in six-well plates. After 24 h, when the cell growth was about 90–95%, the total RNA was extracted and reverse transcribed, and qRT-PCR was performed. The relative mRNA expression of *KLF6*-KO-BMECs showed that the *OXSM, FASN, MCAT, ABCG1,* and *GPAM* were significantly up-regulated in *KLF6-KO-BMECs* as compared with the normal BMECs, which is shown in Figure 16A. Meanwhile, the mRNA expression of *KLF6*-KO-BMECs showed that the expression of the *CBP4, HSD17B8, ACACA, ACACB,* and *AGPA* was significantly down-regulated in *KLF6*-KO-BMECs as compared with the normal BMECs with 4-fold change, as shown in Figure 16B. These results strongly suggested that *KLF6* plays an important role in lipid synthesis by targeting the marker genes related to lipid synthesis.

### 3.16. String Interaction Network of Different Marker Genes of Lipogenesis in Bos Taurus

The string interaction shows that the different genes are interlinked with each other. Figure 17A string interaction showed that (https://stringdb.org/cgi/network?taskId=bNDo5kjnEDxr&sessionId=b8xaTD42Ytio (accessed on 10 February 2022)) the *OXSM* was regulated through the interaction of the *FASN, MCAT*, and *CBP4* genes. Meanwhile, the main enriched KEGG pathways of *OXSM* showed that it regulates the fatty acid metabolism, biosynthesis, biotin metabolism, and biosynthesis of cofactors. In addition, the qPCR results showed that the *KLF6* highly influences these genes and regulates these functions. Figure 17B of string interaction showed that (https://www.stringdb.org/cgi/network?taskId=b062fojmS7GI&sessionId=bw1BF3kKjtMh (accessed on 10 February 2022)) the *SCD* was regulated through the exchange of the *ACACA, FASN,* and *HSD17B12* genes. Meanwhile, the main enriched KEGG pathways of *SCD* showed that it involves regulating the biogenesis of unsaturated fatty acid, metabolic pathway, fatty acid metabolism, and the *PPAR* and *AMPK* signaling pathway. The qPCR results showed that *KLF6* highly influences these genes and regulates these functions. This string interaction and qPCR strongly suggested that *KLF6* regulates the different lipogenesis pathways by targeting different genes.

## 4. Discussion

*Role of bta-miR-148a in milk fat synthesis in BMECs:* With the advancement of biotechnology, numerous researchers have validated the pivotal role of miRNAs in cattle, which are involved with protein synthesis, mammary gland growth, and milk fat synthesis [45]. We aimed to investigate the role of bta-miR-148a and *KLF6* in milk fat metabolism, primarily on the contents of TG, and CHO in BMECs. To this end, the BMECs were transfected with bta-miR-148a-mimic, inhibitor, and shNC. The presence of binding sites in 3’UTR of *KLF6* mRNA that was found through bioinformatic analysis and verified through dual luciferase assay justified the negative correlation between bta-miR-148a and *KLF6*. To further explore the effects of bta-miR-148a on lipid metabolism in BMECs, the BMECs were transfected with bta-miR-148a mimic, inhibitor, and shNC. The results show that bta-miR-148a negatively regulates the expression of *KLF6*. The effect of bta-miR-148a transfection on TG and CHO contents in three experimental groups indicated that TG and CHO contents were low in the bta-miR-148a mimic group as compared to the inhibitor the and control group, while the TG and CHO content in the inhibitor group was high compared to the mimics and control group. Our result was similar to the study mentioned earlier conducted on mice: the deletion of the miR-148 harms hepatic cancer and lipid metabolism by targeting different genes, including *ABCA1,* and *PGC1α* [46]. When the mouse with KO miR-148a germline was used, it was revealed that miR-148a plays an important function in hepatic metabolism [17]. Another research identified that miR-148a has a significant role in immunity and epigenetic regulation [17,26]. The miR-148a, miR-186, and miR-200a are three of the most important curial miRNAs that play a crucial role in lactation and milk yield [17]. As a consequence of the association of miR-148 with the transcriptional factor *MAFB*, miR-148a has a key role in the differentiation of monocytes into osteoclasts [20]. In another investigation, it was shown that miR-148a, through inhibiting the *DNMT1*, plays an important role in the differentiation and maturation of liver cells [21]. Simultaneously, another study suggested that the SNP of miR-148 is linked with obesity. Previous research on marsupial tammar wallaby validated the most copious miRNAs during the lactation stage in the tammar wallaby, including miRNAs of the let-7 family (7f, 7a, and 7i), miR-148, miR-181, miR-184, miR-191, and miR-375 [47]. To this end, the results of this study coincide with the previous research and provide additional information which validates the role of bta-miR-148a in negatively regulating the TG and CHO content in milk fat synthesis in BMECs.

*Role of KLF6 in milk fat synthesis in BMECs:* To delineate the role of *KLF6*, the *KLF6* overexpression, and down-regulation was achieved through the transfection of BMECs with pBI-CMV3-*KLF6* and pb7sk-*KLF6*-siRNA1, respectively. The result of *KLF6* overexpression in BMECs through the transfection of pBI-CMV3-*KLF6* showed a positive correlation with TG and CHO contents. This explains that through the overexpression of *KLF6* in BMECs, the contents of TG and CHO were increased compared to the control group. For cross-checking, the *KLF6* was down-regulated in BMECs through transfection with pb7sk-*KLF6*-siRNA1. The results showed that through the interference of *KLF6* expression, the contents of TG, and CHO were decreased compared to the control group. Overall, these results highlight the importance of *KLF6* in the synthesis of TG and CHO contents in BMECs, which can be generalized to the role of *KLF6* in milk fat metabolism. *KLF6* promotes the development of pre-adipocytes into adipocytes by modulating the expression of delta-like non-canonical notch ligand 1 (DLK1), which is a gene that suppresses adipocyte differentiation according to our results. The overexpression of *KLF6* promoted adipocyte differentiation, while its silencing inhibited adipogenesis in 3T3-L1 cells [48]. Meanwhile, the role of the *KLF6* gene in beef cattle study suggested that the *KLF6* gene might be exploited as a viable marker gene for improving beef breeds through the marker-assisted selection of Qinchuan cattle [49]. In this study, the researcher validated that *KLF6* plays a milestone role in enhancing meat quality in Qinchuan cattle [49]. According to another study, a single *KLF6* allele reduced the risk of prostate cancer by 77% in human subjects highlighting the tumor suppressor attribution of *KLF6* [50]. Meanwhile, another study reported that the amplification of the second exon of the *KLF6* gene reveals the three SNP loci at 3332C > G; 3413C > T, and 3521G > A, which were found to be linked with more excellent body and carcass measurements in cattle [49]. Another study also reported the role of *KLF6* in hepatic FA and lipid metabolism by inducing the expression of *PPARG* and TG accumulation in HepG2 cells. The *KLF6* expression and TG accumulation were increased when HepG2 cells were treated with palmitic acid for imitating conditions such as a fatty liver, while the reverse happened upon the silencing of *KLF6* [51]. The results of this study align with previous studies for the increased contents of TG in BMECs with the forced expression of *KLF6* and decreased contents of TG in BMECs with the silencing of *KLF6*.

*KLF6 target the PPARA and PPARG:* We determined whether *KLF6* affects the *AMPK/mTOR/PPARG* pathway. The qPCR and WB results elucidated the expression of the AMPK up-regulated in the *KLF6*-KO-BMECs compared to the normal BMECs. Meanwhile, the expression of the *mTOR/PPARG* was decreased in the knock-out BMECs compared to the normal BMECs. From qPCR and Western blot, it confirmed that KLF6 through the AMPK/mTOR/PPARG pathway plays a significant role in the lipid synthesis in the BMECs. These results are also similar to the KEEG *AMPK/mTOR/PPARG* pathway. The previous study on *PPARG* reported that in early fat differentiation, *PPARG* is used as a master gene and plays a significant role in fat differentiation [52]. According to other studies in mice, *PPARG* isoform 2 is critical for hepatocyte lipid accumulation and controls the expression of lipogenesis and adipogenesis associated genes [53]. *KLF6* activates the *PPARG* gene, which increases the amount of TG in the body [51,54]. The *PPARG* gene is a marker gene that plays an important role in early fat differentiation [52,55]. So, the previous research validates that *PPARG* is a key pathway that governs lipogenesis. The results of this study also corroborate with previous studies as the *PPARG* expression was significantly down-regulated in the *KLF6*-KO-BMECs compared to the normal BMECs. Overall, these results highlight the importance of the *KLF6* gene in fat synthesis through *PPARG* pathway.

In contrast to the normal BMECs, the *KLF6* knock-out cell line showed a markedly reduced expression of *PPARA* and their selected genes in terms of mRNA and protein, suggesting that these genes are involved in lipogenesis and cholesterol metabolism. Preliminary studies have demonstrated an important role for *PPAR* in controlling the transcriptional activity of several key adipocyte-related genes, including *aP2, C/EBP, GLUT4*, and perilipin in response to insulin [56,57,58,59,60,61,62]. Meanwhile, another study in the mitochondria reported that *PPARA* regulates the fatty acid in the muscle [63,64], and the liver regulates fatty acid by targeting the a-carnitine palmitoyl transferase I genes [65]. The other study elucidated whether the *PPARA*-deficient mice fed with high fat accumulated the lipid in the liver, which showed that the *PPARA* has a major role in lipid metabolism [66]. So, the previous research validates that *PPARA* is a marker pathway that regulates lipogenesis. The findings of this study also coincide with previous studies as the expression of *PPARA* and its related genes was considerably down-regulated in *KLF6*-KO-BMECs as compared to the normal BMECs. During the conduct of this research, the role of the *KLF6* gene in fat synthesis was demonstrated, which is controlled by the *PPARA* and *PPARG* pathways.

The *KLF6* regulates the *PPARA* and *PPARG* pathways, and the other marker genes significantly regulate fat synthesis. The qPCR of the marker genes elucidates the relative mRNA expression of *KLF6*-KO-BMECs and showed that the expression of *OXSM, FASN, MCAT, ABCG1,* and *GPAM* was significantly up-regulated in *KLF6*-KO-BMECs as compared with the normal BMECs. Meanwhile, the relative mRNA expression of *KLF6*-KO-BMECs showed that the expression of *CBP4, HSD17B8, ACACA, ACACB,* and *AGPA* was significantly down-regulated in *KLF6*-KO-BMECs as compared with the normal BMECs. The string interaction validates that these genes are interlinked with each other for the regulation of the function. The research results obtained via genome-wide-association, functional genomics, and comparative genomics analyses indicate that genes including *SCD*, *DGAT1*, *ABCG2*, and *FASN* were reported as genetic markers and candidate genes in milk fat traits [67]. Target interruption of the *LXRα* gene in mice reported lacking the expression of many fatty acid genes, for example, *ACC, FAS*, *SREBP-1c*, and *SCD-1* [68]. Meanwhile, another study reported that the *GPAM* gene has an important role in enhancing the triglyceride content in the BFF cell line [69]. Further study validates that the GPAM gene has an important role in enhancing the triglyceride content in bovine mammary epithelial cells [70]. In contrast, the study on the *ACACB* gene reported that the *SNP* of the *ACACB* gene is associated with fatty acid regulation in milk [71]. Moreover, the study on *ACACA* elucidates that ACACA has increased the milk yield in dairy cattle [72]. Another study reported that the *HSD17B8* gene has a key role in improving beef quality. The previous research validates the role of these genes in fat synthesis and lipid metabolism. Further, string interaction and the KEEG pathway validated that *KLF6* is the major gene that regulates lipid synthesis in BMECs by targeting the *PPARG*, *PPARA*, and their other marker genes of fat synthesis. In this regard, this study provides additional information for validating the Importance of the *PPARA* and *PPARG* genes in lipid synthesis. The KEGG, previous research, and string interaction validate *KLF6*, by targeting *PPARA* and *PPARG* pathways, and the other fat marker genes, regulate fat synthesis in BMECs. Thus, in light of the previous research, in silico analysis, and experimental results of this study, we conclude that *KLF6* is a potential candidate for increased milk and the synthesis of valuable milk components in dairy cattle through marker-assisted selection in molecular breeding. Interestingly, we have also revealed the role of bta-miR-148a and its target gene, *KLF6*, in the lipid metabolism of BMECs. However, further validation of these findings through in vivo experimentation would help the animal breeder's selective breeding. These results could be accounted for to improve the quality of dairy milk and the selection and breeding of dairy cows with the potential to produce better-quality milk.

## 5. Conclusions

We identified that bta-miR-148a negatively regulates the TG and CHO accumulation in BMECs. The *KLF6* gene is a novel direct target of bta-miR-148a, which plays an important role in lipid metabolism in BMECs. The expression of *KLF6* is directly linked to the contents of TG and CHO in BMECs. The bioinformatic and in vitro experimental results validated that *KLF6*, by targeting *PPARA,* the *PPARG* pathway, and the other fat marker genes, regulate fat synthesis in BMECs. Thus, in light of the results, we conclude that bta-miR-148a and *KLF6* are potential candidates for increased milk and the synthesis of valuable milk components in dairy cattle through marker-assisted selection in molecular breeding.

## Figures and Tables

**Figure 1 genes-13-01828-f001:**
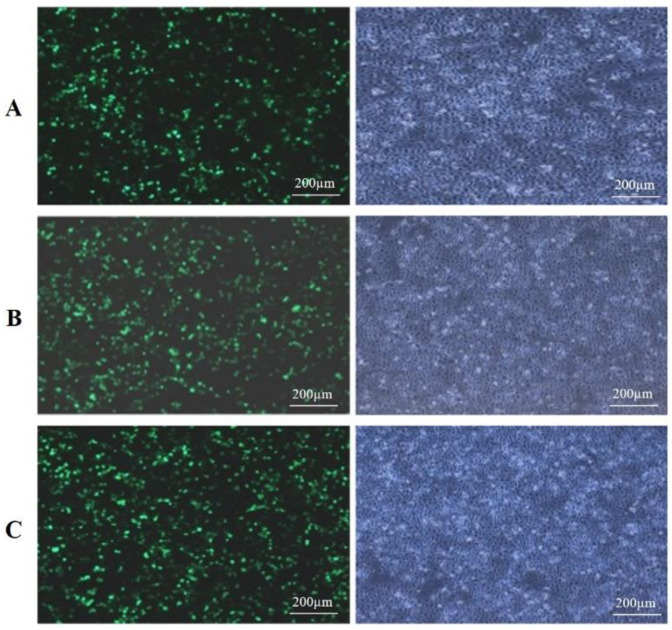
Cell morphology (**right side**) and GFP expression (**left side**) in BMECs under fluorescence microscope after 24 h of transfection with (**A**), bta-miR-148a-mimic; (**B**), bta-miR-148a-inhibitor; and (**C**), bta-miR-148a-shNC.

**Figure 2 genes-13-01828-f002:**
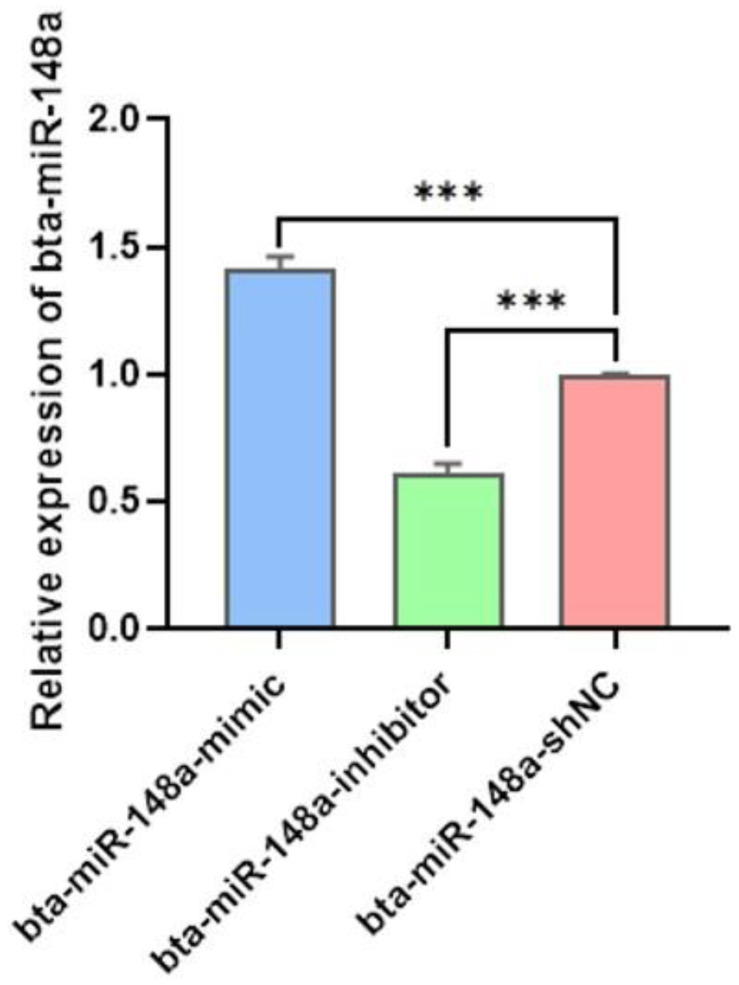
Relative expression of bta-miR-148a in BMECs after 24 h of transfection with bta-miR-148-mimic, bta-miR-148a-inhibitor, and bta-miR-148a-shNC. *** *p* < 0.001.

**Figure 3 genes-13-01828-f003:**
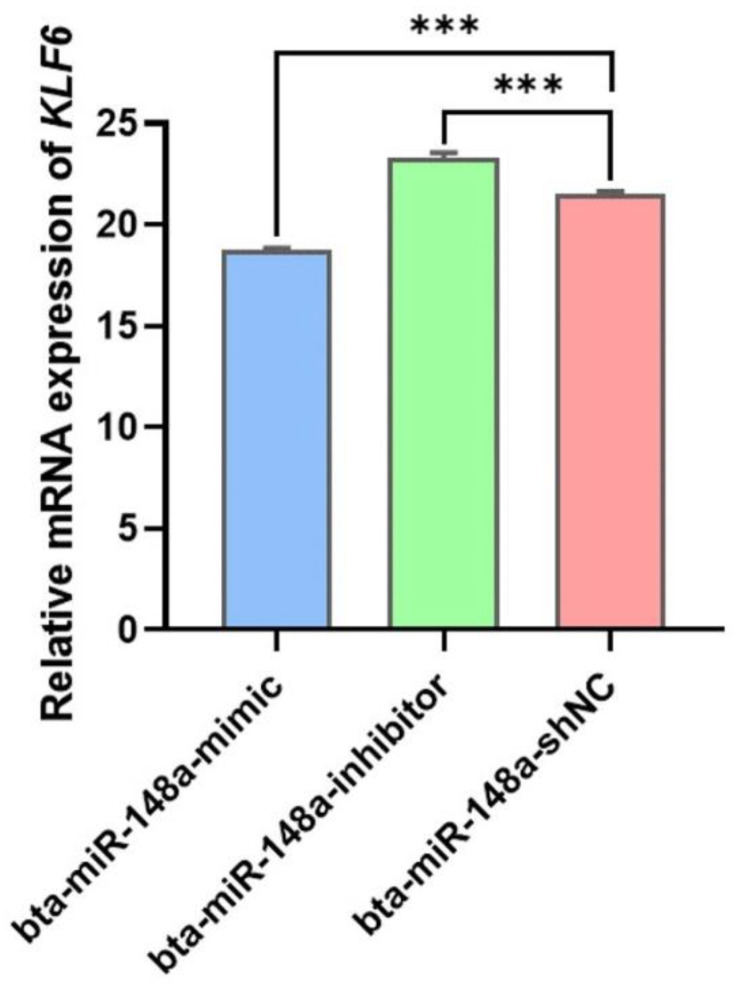
Relative expression of *KLF6* mRNA in BMECs transfected with bta-miR-148a-mimic, bta-miR-148a-inhibitor, and bta-miR-148a-shNC. *** *p* < 0.001.

**Figure 4 genes-13-01828-f004:**
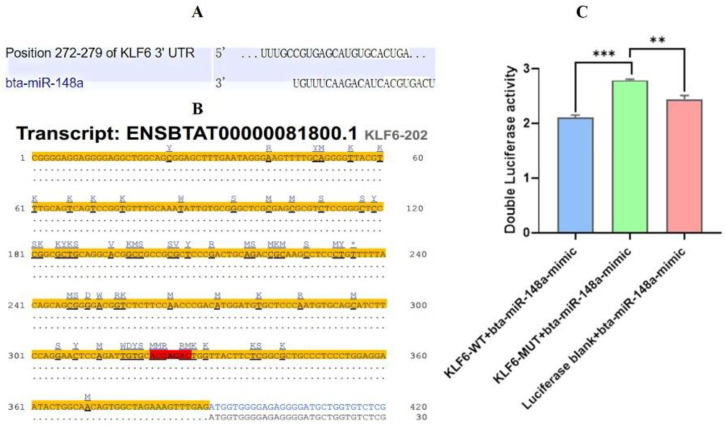
Bioinformatics analysis and dual-luciferase reporter assay. (**A**), Binding site matching in white highlighted area for bta-miR-148a and 3’-UTR KLF6 mRNA. (**B**), Binding site matching results in 3’-UTR *KLF6* mRNA for bta-miR-148a highlighted in red. The yellow highlighted sequence represents the 3’-UTR of *KLF6* mRNA. (**C**); Dual-luciferase reporter assay representing the target site validation. ** *p* < 0.01, *** *p* < 0.001.

**Figure 5 genes-13-01828-f005:**
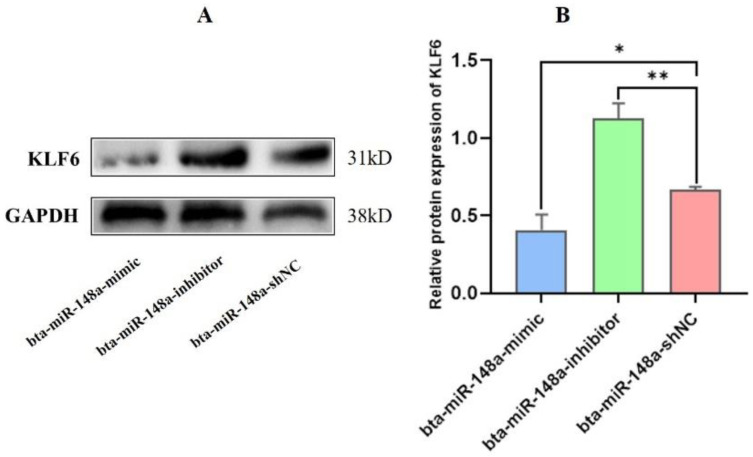
Relative expression of KLF6 protein in BMECs transfected with bta-miR-148a-mimic, bta-miR-148a-inhibitor, and bta-miR-148a-shNC. (**A**); Western blot bands for KLF6 protein in different experimental groups. (**B**); Quantification of KLF6 protein expression. * *p* < 0.05, ** *p* < 0.01.

**Figure 6 genes-13-01828-f006:**
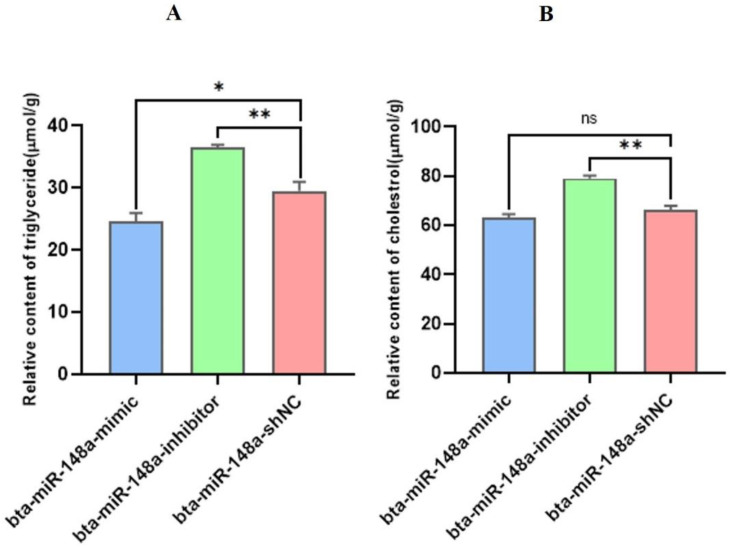
(**A**) TG’s relative contents in BMECs after 24 h of transfection with bta-miR-148a-mimic, inhibitor, and shNC. (**B**) Relative contents of CHO in BMECs after 24 h of transfection with bta-miR-148a-mimic, inhibitor, and shNC. ns = non-significant * *p* < 0.05, ** *p* < 0.01.

**Figure 7 genes-13-01828-f007:**
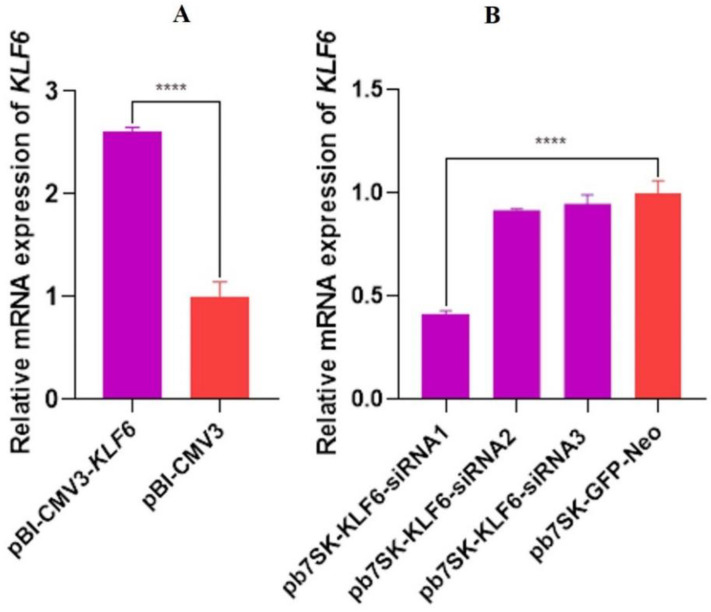
(**A**) Relative expression of *KLF6* mRNA in BMECs transfected with pBI-CMV3-*KLF6* and pBI-GFP-Neo-CMV3. (**B**) Relative expression of *KLF6* mRNA in BMECs transfected with pb7SK-*KLF6*-siRNA1, pb7SK-*KLF6*-siRNA2, pb7SK-*KLF6*-siRNA3, and pb7SK-GFP-Neo. **** *p* < 0.0001.

**Figure 8 genes-13-01828-f008:**
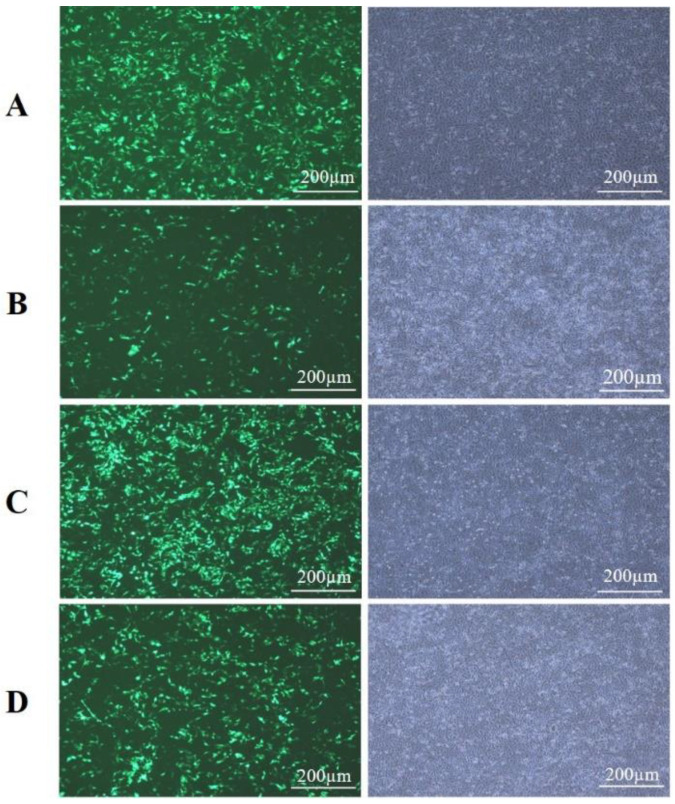
Cell morphology and GFP expression in BMECs under a fluorescence microscope after 24 h of transfection with; (**A**) pBI-CMV3-*KLF6*, (**B**) pBI-GFP-Neo-CMV3, (**C**) pb7sk-*KLF6*-siRNA1, (**D**) pb7sk-GFP-Neo.

**Figure 9 genes-13-01828-f009:**
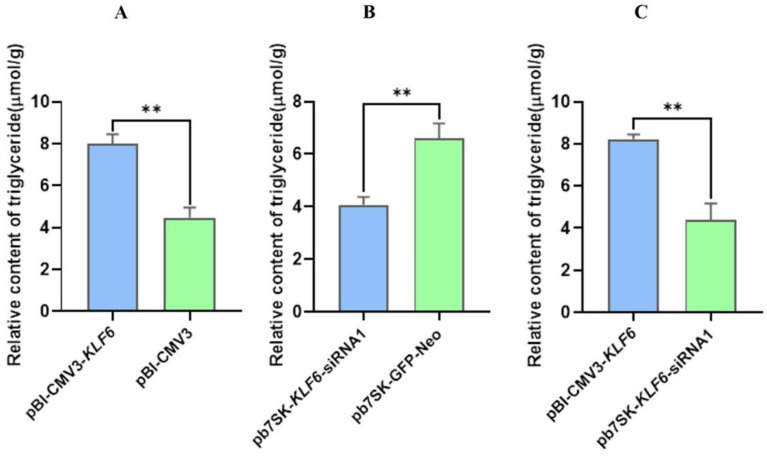
(**A**) Relative contents of TG in BMECs transfected with pBI-CMV3-*KLF6* and pBI-GFP-Neo-CMV3. (**B**) Relative contents of TG in BMECs transfected with pb7SK-*KLF6*-siRNA1 and pb7SK-GFP-Neo. (**C**) Relative contents of TG in BMECs transfected with pBI-CMV3-*KLF6* and pb7SK-*KLF6*-siRNA1. ** *p* < 0.01.

**Figure 10 genes-13-01828-f010:**
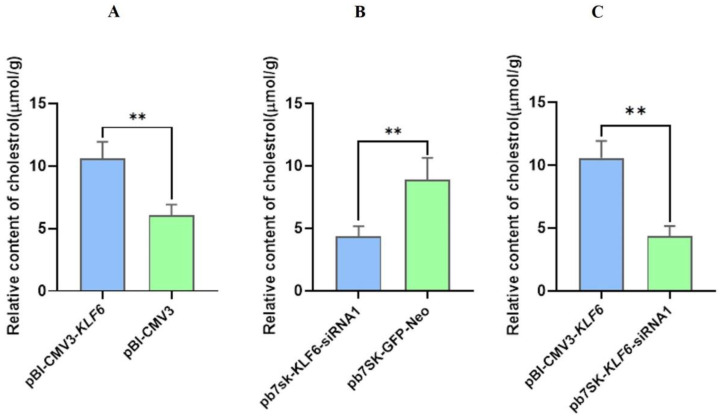
Cholesterol content in BMECs after *KLF6* overexpression and down-regulation (**A**) Relative contents of cholesterol in BMECs transfected with pBI-CMV3-*KLF6* and pBI-GFP-Neo-CMV3. (**B**) Relative contents of cholesterol in BMECs transfected with pb7SK-*KLF6*-siRNA1 and pb7SK-GFP-Neo. (**C**) Relative contents of cholesterol in BMECs transfected with pBI-CMV3-*KLF6* and pb7SK-*KLF6*-siRNA1. ** *p* < 0.01.

**Figure 11 genes-13-01828-f011:**
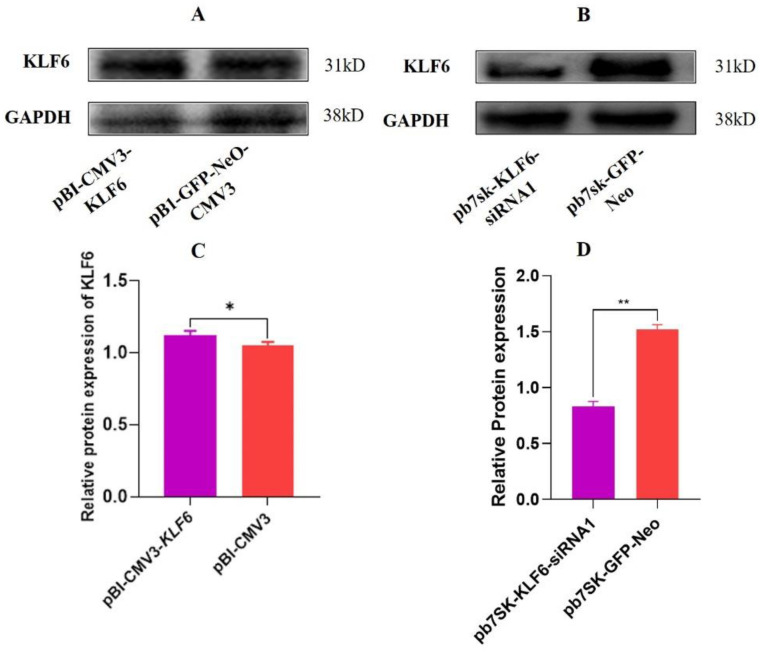
Relative protein expression of KLF6 in BMECs (**A**) Relative protein expression of KLF6 in BMECs transfected with pBI-*KLF6*-CMV3, pBI-GFP-Neo-CMV3. (**B**) Relative protein expression of KLF6 in BMECs transfected with pb7sk-*KLF6*-siRNA1, pb7sk-GFP-Neo. (**C**) Relative protein expression of KLF6. (**D**) Relative protein expression of KLF6. * *p* < 0.05, ** *p* < 0.01.

**Figure 12 genes-13-01828-f012:**
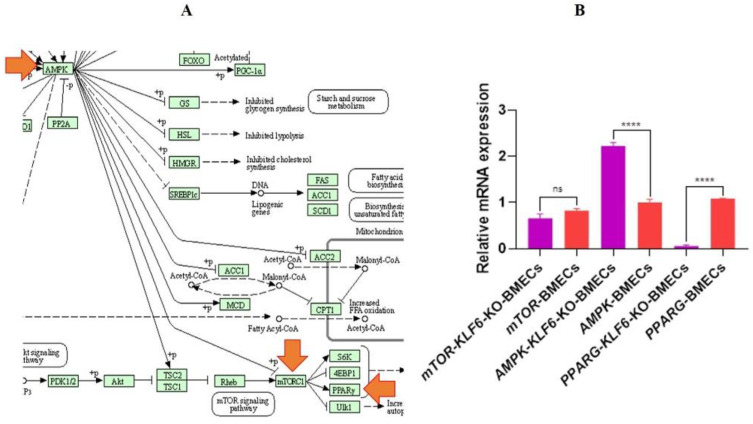
Analysis of lipid metabolism-related genes (**A**) The KEGG pathway of *AMPK/mTOR/PPARG.* Arrows point out the lipid metabolism-related pathways (**B**) The relative mRNA expression of *AMPK/mTOR/PPARG* in the *KLF6*-KO-BMECs and normal BMECs. ns = non-significant, **** *p <* 0.0001.

**Figure 13 genes-13-01828-f013:**
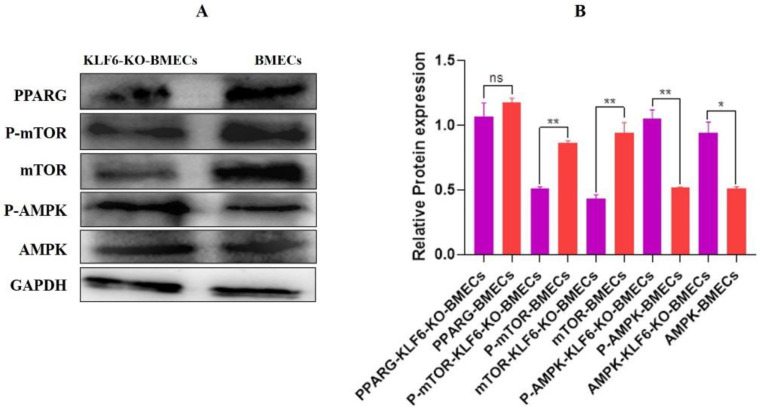
Protein expression of lipid metabolism-related genes (**A**,**B**) The relative protein expression of AMPK/mTOR/PPARG in the *KLF6*-KO-BMECs and normal BMECs. * *p* < 0.05, ** *p* < 0.01.

**Figure 14 genes-13-01828-f014:**
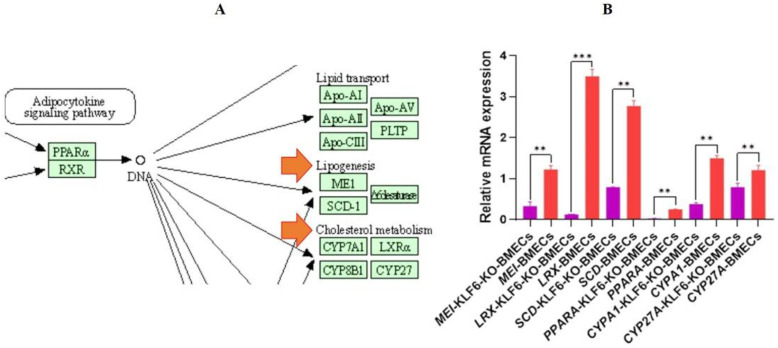
Analysis of lipid metabolism-related genes (**A**) The KEGG pathway of *PPARA.* Arrows points out the *PPARA* pathways genes associated with lipogenesis and cholesterol metabolism *(***B**) The relative mRNA expression of *PPARA* and its pathway related genes in the *KLF6*-KO-BMECs and normal BMECs. ** *p* < 0.01, *** *p* < 0.001.

**Figure 15 genes-13-01828-f015:**
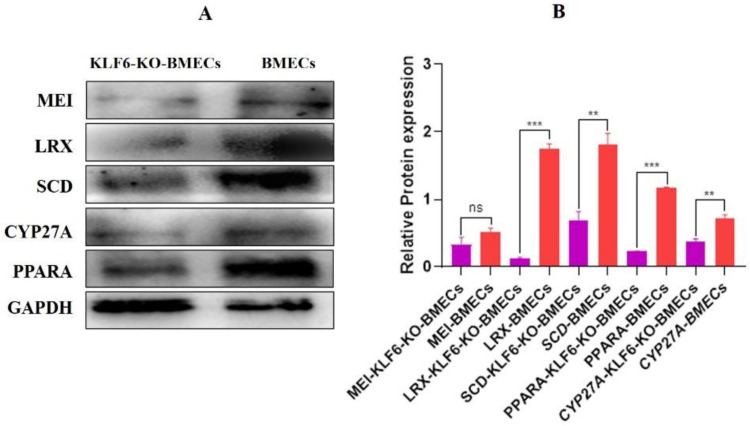
Protein expression of lipid metabolism-related genes (**A**,**B**) The relative protein expression of PPARA and their genes are associated with lipogenesis and cholesterol metabolism in the *KLF6*-KO-BMECs and normal BMECs. ns = non-significant, ** *p* < 0.01, *** *p* < 0.001.

**Figure 16 genes-13-01828-f016:**
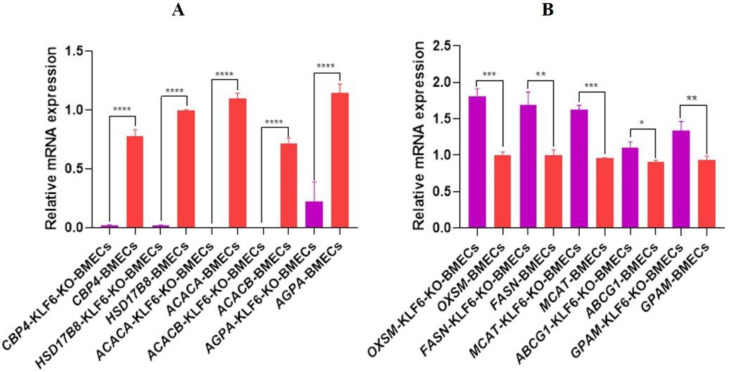
(**A**,**B**); mRNA expression of marker genes associated with lipogenesis in the *KLF6*-KO-BMECs and normal BMECs. * *p* < 0.05, ** *p* < 0.01, *** *p* < 0.001, **** *p* < 0.0001.

**Figure 17 genes-13-01828-f017:**
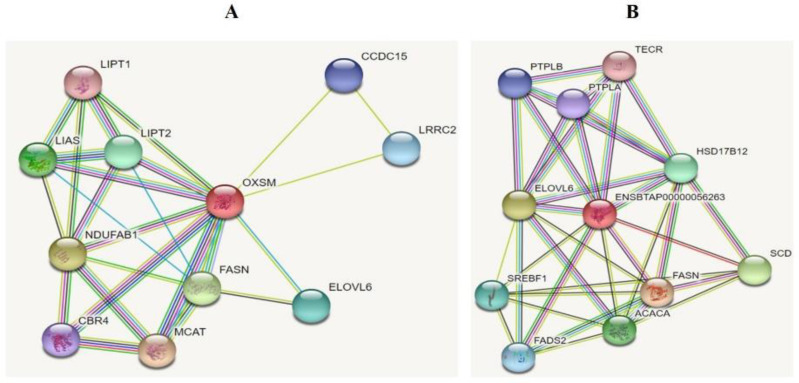
(**A**,**B**) String interaction of different fat metabolism-related genes of Bos Taurus.

**Table 1 genes-13-01828-t001:** Real-time PCR primer sequences.

Name	Primer	Sequence (5’–3’)
Bta- miR148a U6 *KLF6* *GAPDH*	F RT F R F R F R	TGCGGTTCAGTGCACTACAGAA GTCGTATCCAGTGCAGGGTCCGAGGTGCACTGGATACGACACAAGTT CTCGCTTCGGCAGCACA AACGCTTCACGAATTTGCGT CTCGAGTCTAGCTGTTAATGCACTGTAGATCTTAATAAT GGCCGCATTATTAAGATCTACAGTGCATTAACAGCTAGA GCATCGTGGAGGGACTTATGA GGGCCATCCACAGTCTTCTG

**Table 2 genes-13-01828-t002:** Primer sequences for dual-luciferase reporter assay.

Name	Primer	Sequence (5’–3’)
*KLF6*-WT	F R	GGCCTCTTCCAGGAACTCCAGATTGTGCACGAGACTGGTTACTTCTCGGCGCTGCCCTC TCGAGAGGGCAGCGCCGAGAAGTAACCAGTCTCGTGCACAATCTGGAGTTCCTGGAAGA
*KLF6*-MUT	F R	GGCCTCTTCCAGGAACTCCAGATTGTGCGATCGACTGGTTACTTCTCGGCGCTGCCCTC TCGAGAGGGCAGCGCCGAGAAGTAACCAGTCGATCGCACAATCTGGAGTTCCTGGAAGA

**Table 3 genes-13-01828-t003:** Small interfering RNA (siRNA) sequences for *KLF6* gene.

Name	Primer	Sequence (5’–3’)
si*KLF6*-1 si*KLF6*-2 si*KLF6*-3	F R F R F R	AGAGGAAGCGATGAGTTAACCAGTTGATATCCGCTGGTTAACTCATCGCTTCTTTTTTG GATCCAAAAAAGAAGCGATGAGTTAACCAGCGGATATCAACTGGTTAACTCATCGCTTC AGAGGCTCCAGATTGTGCACGAGATTGATATCCGTCTCGTGCACAATCTGGAGTTTTTTG GATCCAAAAAACTCCAGATTGTGCACGAGACGGATATCAATCTCGTGCACAATCTGGAGC AGAGGTCTGAGCTCCTCGGTCACCTTGATATCCGGGTGACCGAGGAGCTCAGATTTTTTG GATCCAAAAAATCTGAGCTCCTCGGTCACCCGGATATCAAGGTGACCGAGGAGCTCAGAC

**Table 4 genes-13-01828-t004:** Real-time PCR primer sequences of *PPARG* and *PPARA* pathway related, and other marker genes of lipid synthesis.

Gene	Primer F (5’–3’)	Primer R (3’–5’)
Bos-*CBR4*	TAGGTCGCGTTAATTTCTTGGT	TCAGGGCAGCTCTACAGGTC
Bos-*OXSM*	CTGCTTACGTGCCAAGAGGT	GTGGGAGGAGACATGGACTTG
Bos-*ACSF3*	ACCTCTACTTGCGCAGCCT	CACAAAGGAGACGTCGTTGG
Bos-*MCAT*	CAGGTAGTGGGCATGGGC	CAGCTCCAGCAGGTCGTAG
Bos-*HSD17B8*	CGCCGTCTGTCGTTGTGTC	ACTTTGTCCCAGTTGTCCTCAG
Bos-*ACSL5*	TGCTGTGTCTGACAATGGGC	TGCTCGATCAGACACCTGTT
Bos-*ACSL1*	AGTGATGGTGCCCGGAGAT	TAGGGTTGGTCTGGTTTCCG
Bos-*FASN*	TCACCTACGAGGCCATTGTG	CTGAAGCCTCAGAGCCACTC
Bos-*ACACB*	CACCTCTGCCACAGAATCCC	GTGCCTGCTTCCTGTCTTCT
Bos-*ACACA*	CACTGTAGCCTCTCCAGCAG	CCATTGTTGGCAATGAGAACCT
Bos-*LXRα*	TCAACCCCATCTTCGAGTTC	ACGACTACTTTGACCACTCG
Bos-*ABCG1*	GACTCGGTCCTCACGCAC	CGGAGAAACACGCTCATCTC
Bos-*SREBP1*	ATGCCATCGAGAAACGCTAC	CTCTTGGACTCAGACGCCTG
Bos-*PPARG*	CGTGGACCTTTCTATGATGGATG	GATACAGGCTCCACTTTGATTGC
Bos-*GPAM*	ATTGACCCTTGGCACGATAG	AACAGCACCTTCCCACAAAG
Bos-*AGPAT6*	AAGCAAGTTGCCCATCCTCA	AGCTTTAACCTCGGTGTCAAA
Bos-*IDH1*	CGATGAGAAGAGAGTGGAGGA	CAAGCCGGGGTATATTTTTG
*Bos-CYP7A1*	GGAAGCGGTACCTGGATGGC	CCCCTGGGGTCTCAGGACAA
*Bos-CYP27A1*	GGAAGCGGTACCTGGATGGC	CAAGGCCGCCTGGATCTCTG
*Bos-ME1*	CAAGGAGCTGGAGAGGCTGC	AGAACGCACCACCAATCGCA
*Bos-GAPDH*	GCATCGTGGAGGGACTTATGA	GGGCCATCCACAGTCTTCTG

## Data Availability

This study does not report any new omics data to include or deposit in the archives.

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
