# Peer review of "Deciphering the Key Regulatory Roles of KLF6 and Bta-miR-148a on Milk Fat Metabolism in Bovine Mammary Epithelial Cells"

_genes, 2022, doi:10.3390/genes13101828_

Round 1
Reviewer 1 Report
In the manuscript entitled “Bta-miR-148a regulates milk fat metabolism in bovine mammary epithelial cells by targeting Kruppel-Like factors 6 gene”, the authors investigated the influence of miR-148a and KLF6 in bovine mammary epithelial cells (BMEC).
The study is interesting and provides evidence of the effects of miR-148a on the expression of genes involved in lipid metabolism in BMEC and therefore on milk fat synthesis. The work was well-done but the manuscript needs some modifications.
The general remarks are:
· The structure of the manuscript could be simplified, in particular with the division of paragraphs 3.12 to 3.16 could be revised. This separation of the different paths led to repeat the experimental conditions. The article is complicated to follow.
· The conclusion of some paragraphs is too hasty. The authors need to be more careful. The results suggest but not demonstrate certain assertions. In addition, the study was performed in vitro needs validation in vivo study. This point must be mentioned in the discussion part.
· Correlation analyses between miR-148a and the expression of candidate genes including KLF6 would be more convincing.
· Some results not in agreement must be discussed: for instance: line 479: KLF6 targets PPARG pathway, but PPARG is not changed at proteic level.
· The manuscript must be check:
o Some sentences lack a verb.
o Is the author was alone to perform the work: My results (e.g.: line 632, 647, 676,…)
o Gene and miRNA names need to be written in italic.
o Use of abbreviations should be checked: sometime CHO something cholesterol in the same sentence. HDF not explained etc…
The specific remarks are:
· Revise sentences:
o line 134: by (Genewiz;
o lines 134-37
o line 139 with (Takara
o line 201
· Check the name of companies for example: is it (TIANGEN BIOTEH ….) or (Tiangen biotech…); line 276; line 278
· line 158: remove space to “bta -miR-148a”
· Due to the large number of illustrations, Table 4 could be in supplemental material
· Line 231: usually tRNA is used for transfer RNA. Here it is probably total RNA
· Line 344: bioinformatics has no effects
· Line 378 & 383: why “respectively”
· Line422: with a p<0.01 why it is only suggestively. Whereas some conclusions are hasty here, the author might write “was higher”
· Line 461: replace “while” by “While”
· Line 485: revise the title of 3.13: not only PPARA
Thus, the work is complete and well done but the manuscript needs to be revised.

Author Response
Reviewer 1
Comments and Suggestions for Authors
In the manuscript entitled “Bta-miR-148a regulates milk fat metabolism in bovine mammary epithelial cells by targeting Kruppel-Like factors 6 gene”, the authors investigated the influence of miR-148a and KLF6 in bovine mammary epithelial cells (BMEC).
The study is interesting and provides evidence of the effects of miR-148a on the expression of genes involved in lipid metabolism in BMEC and therefore on milk fat synthesis. The work was well-done but the manuscript needs some modifications.
The general remarks are:
- The structure of the manuscript could be simplified, in particular with the division of paragraphs 3.12 to 3.16 could be revised. This separation of the different paths led to repeat the experimental conditions. The article is complicated to follow.
- The conclusion of some paragraphs is too hasty. The authors need to be more careful. The results suggest but not demonstrate certain assertions. In addition, the study was performed in vitroneeds validation in vivo study. This point must be mentioned in the discussion part.
Ans: Dear reviewer, thank you for your valuable suggestion. We have revised the certain assertions into suggestive implication of the findings and also added the need for in vivo study demonstration.
- Correlation analyses between miR-148aand the expression of candidate genes including KLF6 would be more convincing.
Ans: Thank you for your suggestion. Over a dozen target genes of bta-miR-148a linked to lipid metabolism were screened in addition to KLF6 via bioinformatics prediction. From these target genes predicted in silico, we selected KLF6 as a candidate target gene for finding its role in milk fat metabolism in BMECs of Chinese Holstein cattle. In the 3'UTR of KLF6 gene mRNA, the bta-miR-148a binding site predicted in silico (Fig.4 A-B) was further validated by the renilla luciferase assay. In addition to the mRNA and protein expression of KLF6 and luciferase reporter assay, revealed that KLF6 is negatively regulated by bta-miR 148a. Which validates that the KLF6 is the target gene of bta-miR-148a. For the identification the role of the KLF6 gene in construct the overexpression vector and silencing vector. The mRNA and protein expression of KLF6 overexpression and silencing was determined through qPCR and WB. While the pathway PPARA, PPARG and all the other fat metabolism genes in the KO-KLF6-MEC was down regulated as compared to the normal mammary epithelial cells. Which strongly validates that bta-miR-148a target the KLF6. The KLF6 through PPARA, PPARG, and the other genes that have important role in milk fat metabolism in mammary epithelial cells. All of these results revealed that the KLF6 and the other genes are potential candidate genes for the bta-miR-148a.
Some results not in agreement must be discussed: for instance: line 479: KLF6 targets PPARG pathway, but PPARG is not changed at protein level.
Ans: Dear reviewer, thank you for your valuable suggestion. The protein expression of PPARG was down-regulated in the KO-KLF6-BMEC as compared to the normal cell line. The protein expression of PPARG is not significantly but the trend is same like mRNA expression.
- The manuscript must be check:
- Some sentences lack a verb.
Ans: Thank you for noting this deficiency, we have improved this in the revised version of manuscript.
- Is the author was alone to perform the work: My results (g.: line 632, 647, 676,…)
Ans: Mistakenly some errors remained in the last version. We have revised those sentences.
- Gene and miRNA names need to be written in italic.
Ans: Dear reviewer, thank you for mentioning, we have revised the gene names into italics, but for the miRNA, the mature sequences like the plasmids are not given in italic.
- Use of abbreviations should be checked: sometime CHO something cholesterol in the same sentence. HDF not explained etc…
Ans: Dear reviewer, thank you for the suggestion. We have tried to correct the abbreviations. However, there is no HDF mentioned in the article.
The specific remarks are:
- Revise sentences:
Ans: Thank you for highlighting these typos. We have revised them all.
o line 134: by (Genewiz;
o lines 134-37
o line 139 with (Takara
o line 201
- Check the name of companies for example: is it (TIANGEN BIOTEH ….) or (Tiangen biotech…); line 276; line 278
- line 158: remove space to “bta -miR-148a”
- Due to the large number of illustrations, Table 4 could be in supplemental material
Ans: Thank you for your suggestion. We would follow the journal production department if they consider this table to be given in supplementary data based on the final drafting of the article.
- Line 231: usually tRNA is used for transfer RNA. Here it is probably total RNA
Ans: Thank you for mentioning it. Yes, it is total RNA and we have revised the tRNA to total RNA.
- Line 344: bioinformatics has no effects
Ans: We have revised this point.
- Line 378 & 383: why “respectively”
Ans: Thank you for pointing it out. We have deleted “respectively” as it was not required in the sentence.
- Line422: with a p<0.01 why it is only suggestively. Whereas some conclusions are hasty here, the author might write “was higher”
Ans: Thank you for commenting it. We have revised this in this in the manuscript.
- Line 461: replace “while” by “While”
Ans: We have revised it. Thank you for pointing it out.
- Line 485: revise the title of 3.13: not only PPARA
Ans: We have revised it. Thank you for pointing it out.
Thus, the work is complete and well done but the manuscript needs to be revised.

Reviewer 2 Report
The work “Bta-miR-148a regulates milk fat metabolism in bovine mammary epithelial cells by targeting Kruppel-Like factors 6 gene” is a very interesting manuscript that assess how the KLF6 gene can be regulated in its functions also by miR-148a, playing a potential role in lipid metabolism in BMECs.
The work presented by the authors is quite exhaustive, considering main pre/post transcriptional events involved in TG and CHO accumulation in BMECs. I found it lacks sufficient information regarding gene expression analysis.
As a general comments, I think that the assessment of RNA quality and quantity by only agarose gel electrophoresis and spectrophotometer is not ideal. At least a run on Bionanalyzer to verify RNA integrity (RIN) should be performed.
Regarding gene expression experiments, usually an up regulation of 2 fold (or, on the contrary, a down regulation of 0.5 fold in linear scale) is considered significant to state clear fold changes in considered genes. Also if fixed thresholds for gene expression studies are still a subject of debate in the scientific community, usually, to prove the statistical significance of DEG (differentially expressed genes) with up/down regulations below 1.5-2 fold, an higher number of biological and technical replicates should be considered, in order to report gene expression data with proper confidence intervals (CI).
In several data reported by the authors the recorded fold changes (in which not CI, but only SEM were chosen to report measurement uncertainty) are below that threshold. Moreover the scale of the fold change shown in all figures is not specified (linear scale? log2? Log10?).
Finally, regarding all qPCR assays reported in the text and in the tables, some critical aspects of their design are missing (for example, how primers have been chosen? Are primer designed on spliced junction of exons to prevent amplification of genomic DNA? Have no-RT-controls, conventional positive and negative controls been evaluated? Etc..). I strongly suggest to report in a supplementary section of the manuscript a checklist to prove that the Real time qPCR Assays follows the MIQE guidelines (minimum information for publication of quantitative real-time PCR experiments). See papers of Bustin et al., 2009 and the more recent MIQE revision of 2022. Moreover, a useful alternative to Real Time PCR is represented by digital PCR, that nowadays is becoming quite mandatory to verify changes in low expressed genes like miRNAs and transcriptional factors. Why digital PCR was not considered by the authors?
I suggest the authors to consider all cited aspects of gene expression measurements in the discussion session of their manuscript, in order to scale back their findings or to strengthen the conclusions of their study.
Minor comments:
Line 12: also if already written in the title of manuscript and in the abbreviations list at the end of the manuscript, please specify (only here for the first time) the acronym KLF6 - Krüppel like factor 6
Line 39: please specify here the acronym of TAG.
Line 40: please specify (only the first time) the acronym “wt”
Line 163 and 241: Have been other reference genes been tested during your experiments? Please justify with proper references and/or supplementary data (for example Genorm, normfinder, bestkeeper data) the choice of GAPDH and U6 for normalization of gene expression measurements in your bovine mam-2 mary epithelial cells.
Fig 2, 3 and all the other figures reporting gene expression data measurements: please specify the unit of measure of the relative expression (linear? Log2? Log10?).
Author Response
Reviewer 2
The work “Bta-miR-148a regulates milk fat metabolism in bovine mammary epithelial cells by targeting Kruppel-Like factors 6 gene” is a very interesting manuscript that assess how the KLF6 gene can be regulated in its functions also by miR-148a, playing a potential role in lipid metabolism in BMECs.
The work presented by the authors is quite exhaustive, considering main pre/post transcriptional events involved in TG and CHO accumulation in BMECs. I found it lacks sufficient information regarding gene expression analysis.
As a general comments, I think that the assessment of RNA quality and quantity by only agarose gel electrophoresis and spectrophotometer is not ideal. At least a run on Bionanalyzer to verify RNA integrity (RIN) should be performed.
Ans: Thank you so much for your suggestions you suggestion is very important to improved our paper quality. Due to availability of the gel electrophoresis and spectrophotometer apparatus we can only use this two for checking the quality and quantity of the RNA. In the fallowing table and picture you can check the quality and quantity of the RNA.
Table OD and concentration value of the extracted RNA
|
Name |
OD |
Concentration |
|
Bta-miR-148a-mimic |
1.87 |
442.8 |
|
Bta-miR-148a-inhibitor |
1.84 |
362.1 |
|
Bta-miR-148a-shNC |
1.84 |
409.4 |
This conventional method also effectively and used world-wide.
Regarding gene expression experiments, usually an up regulation of 2 fold (or, on the contrary, a down regulation of 0.5 fold in linear scale) is considered significant to state clear fold changes in considered genes. Also, if fixed thresholds for gene expression studies are still a subject of debate in the scientific community, usually, to prove the statistical significance of DEG (differentially expressed genes) with up/down regulations below 1.5-2-fold, an higher number of biological and technical replicates should be considered, in order to report gene expression data with proper confidence intervals (CI).
Ans: Thank you for your suggestion. The data analysis calculates relative expression using delta/delta Ct method, in which delta Ct is calculated between gene of interest (GOI) and an average of reference genes (HKG), followed by delta-delta Ct calculations (delta Ct (Test group) - delta Ct (control group)). Fold change is then calculated using 2(-delta delta Ct) formula. The classic method is suitable too. Secondly, we must consider the multi-effect of the genes we study, a lot of reference make the screening of genes as more than 1.5-fold change, not 2-fold. For the complex trait we need 1.5-fold if we will.
In several data reported by the authors the recorded fold changes (in which not CI, but only SEM were chosen to report measurement uncertainty) are below that threshold. Moreover the scale of the fold change shown in all figures is not specified (linear scale? log2? Log10?).
Ans: Thank you for your suggestion. The data analysis calculates relative expression using delta/delta Ct method, in which delta Ct is calculated between gene of interest (GOI) and an average of reference genes (HKG), followed by delta-delta Ct calculations (delta Ct (Test group) - delta Ct (control group)). Fold change is then calculated using 2(-delta delta Ct) formula. The classic method is suitable too. Now fold change all picture added.
Finally, regarding all qPCR assays reported in the text and in the tables, some critical aspects of their design are missing (for example, how primers have been chosen? Are primer designed on spliced junction of exons to prevent amplification of genomic DNA? Have no-RT-controls, conventional positive and negative controls been evaluated? Etc..). I strongly suggest to report in a supplementary section of the manuscript a checklist to prove that the Real time qPCR Assays follows the MIQE guidelines (minimum information for publication of quantitative real-time PCR experiments). See papers of Bustin et al., 2009 and the more recent MIQE revision of 2022. Moreover, a useful alternative to Real Time PCR is represented by digital PCR, that nowadays is becoming quite mandatory to verify changes in low expressed genes like miRNAs and transcriptional factors. Why digital PCR was not considered by the authors? I suggest the authors to consider all cited aspects of gene expression measurements in the discussion session of their manuscript, in order to scale back their findings or to strengthen the conclusions of their study.
Ans: Thank you for your suggestion. The sequence of bta- miR-148a was surfed from (http://www.mirbase.org). This study designed primers by Primer Premier 6.0.
Table Reverse transcription reaction mixture
|
Reagents |
Amount |
|
RNA |
1 µg (3/concentration×1000) |
|
5xgDNA wiper mix |
2 µl |
|
RNase -free Water |
to 10 µl |
Put for 2 minutes at 42 degrees centigrade
|
10RT mixture |
2 µl |
|
HiScript II enzyme mixture |
2 µl |
|
148 RT Primer/U6 Primer/ Syber green mixture |
1 µl |
|
RNase -free Water |
5 µl |
|
Total |
20 µl |
The kit was used for the qPCR (Vazyme, CUSqMM, Q711, China). The reaction mixture was prepared in the 96 well plates by the given protocol in table. The real-time quantitative qPCR consists of a two steps experiment. The pre-degeneration process took place at 95 degrees Celsius for 5 minutes. At 95 degrees Celsius for 10 seconds, then at 60 degrees Celsius for 30 seconds, extend the degeneration process for another 30 seconds, totaling 42 cycles. Each cycle rose by 0.5 degrees Celsius. Finally, the temperature rose to 95 degrees Celsius for 15 seconds and was read on the final plate reading.
Real quantitative results are calculated and data analysis of the final values in Microsoft Excel through 2-ΔΔ CT using the GAPDH as a reference gene.
Table shows the qPCR reaction mixture
|
Reagents |
Amount |
|
2×miRNA Universal SYBR qPCR Master Mix |
10 µl |
|
Forward Primer |
0.4 µl |
|
Reversed Primer |
0.4 µl |
|
cDNA |
1 µl |
|
ddH2o |
to 20 µl |
The binding site of the target gene KLF6 at bta-miR-148a through predicated through Target Scan (http://www.targetscan.org). And then used the ensemble software for designing the KLF6-Wild-Type and KLF6-MUTANT primers. While for the KLF6 interference vector construction, two complementary oligos with restriction cutting sites on the 5′ end was designed using the shRNA Construct Builder tool of Gene-Script Biotech and synthesized by Sangon Biotech (Shanghai;). The connected shRNA vectors were sent to the company for sequencing, and the results showed that the sequence matched those shRNAs constructed through the Gene Script shRNA designing tool. This all software and method of qPCR effective too for the identification the expression of the minor gene or miRNA expression.
Minor comments:
Line 12: also if already written in the title of manuscript and in the abbreviations list at the end of the manuscript, please specify (only here for the first time) the acronym KLF6 - Krüppel like factor 6
Ans: Thank you for your suggestion. Now we revised the line 12 according to your suggestion and changing highlight in colour.
Line 39: please specify here the acronym of TAG.
Ans: Thank you for your suggestion. Now we revised the line 39 according to your suggestion and changing highlight in colour.
Line 40: please specify (only the first time) the acronym “wt”
Ans: Thank you for your suggestion. Now we revised the line 40 according to your suggestion and changing highlight in colour.
Line 163 and 241: Have been other reference genes been tested during your experiments? Please justify with proper references and/or supplementary data (for example Genorm, normfinder, bestkeeper data) the choice of GAPDH and U6 for normalization of gene expression measurements in your bovine mam-2 mary epithelial cells.
Ans: Thank you for your suggestion. we used B-actin housekeeping gene for genes. But when we did our experiment with the GAPDH and U6 our experiment system was more stable and significant as compared to the other housekeeping genes. That why we selected these two genes as housekeeping gene in our experiment. However, our previous work on another miRNA proved that U6 is best housekeeping gene for miRNA (MiR-485 targets the DTX4 gene to regulate milk fat synthesis in bovine mammary epithelial cells) While another study which proved that the GAPDH is the best housekeeping gene in mammary epithelial cells (C4BPA: A Novel Co-Regulator of Immunity and Fat Metabolism in the Bovine Mammary Epithelial Cells). Not only our group but other study also proved GAPDH and U6 for normalization of gene expression.
Fig 2, 3 and all the other figures reporting gene expression data measurements: please specify the unit of measure of the relative expression (linear? Log2? Log10?).
Ans: Thank you for your suggestion. The data analysis calculates relative exprssion using delta/delta Ct method, in which delta Ct is calculated between gene of interest (GOI) and an average of reference genes (HKG), followed by delta-delta Ct calculations (delta Ct (Test group) - delta Ct (control group)). Fold change is then calculated using 2(-delta delta Ct) formula. The classic method is suitable too. There no unit, only though the relative expression fold change value to show that. Now we added the fold change in all figures which reporting gene expression.

Reviewer 3 Report
The authors show that miR-148a regulates the abundance of the KLF6 messenger and downstream genes that affect fat metabolism. The work is original and its content is significative. However, the second part of paper are several assays aimed at verifying effects not indicated in the title. Other experiments modifying KFL6 expression by other methods. I suggest that the experiments from the point 3.6 to 3.17 be removed or introduced in a better way.
Till point 3.5, some considerations:
In the table 1, there some primers are too long for real time PCR. Please check them.
In "Results", at the point 3.1, instead of an estimate of the transfection frequency at 24 h, the authors show figures of cells expressing gfp, where we can see that the transfection rate is high, but could be any percentage from 50-100. The transfection frequency helps to interpret the differences observed in the following assays, so I consider it important to know it. If the transfection rate is 50%, the observed differences could be corrected by a factor of two.
The 4A, 4B and 13A figures should be improved
The names of the categories in the figures of relative expressions should be simplified. For example, in stead of bta-miR-148a-mimic, bta-miR-148a-inhibitor and bta-miR-148a-shNC , use only miR-148a, anti miR-148a and shNC. The names are too long, and only a few letters are informative.
The figure 3 is a relative expression. Which expression correspond to expression 1? I don't understand the scale.
Figure 6A and 6B are presented as relative content but with units (umol/ug). Contradictory
I suggest huge correction before published
Author Response
Reviewer 3
The authors show that miR-148a regulates the abundance of the KLF6 messenger and downstream genes that affect fat metabolism. The work is original and its content is significative. However, the second part of paper are several assays aimed at verifying effects not indicated in the title. Other experiments modifying KFL6 expression by other methods. I suggest that the experiments from the point 3.6 to 3.17 be removed or introduced in a better way.
Ans: Thank you so much for your valuable comments it has great importance for us to improved our paper. This research work shows the role of the bta-miR-148a and its target gene in milk fat metabolism. Our purpose of this research not only identify the role of the miR-148a, and KLF6 but we want to elucidate the pathway which KLF6 involved for the regulation of fat in mammary epithelial cells. That why this research work divided into four parts.
1: Identification the role of bta-miR-148a in milk fat
2: Identification the target gene of bta-miR-148a
3: Identification the role of target gene (KLF6) in milk fat
4: Identification the pathway of target gene (KLF6).
However, we revised it and changing highlights in color. We want to clear our results to explain in detailed. But if you want to changed it further, please let us know we will do it.
Till point 3.5, some considerations:
In the table 1, there some primers are too long for real time PCR. Please check them.
Ans: Thank you so much for your suggestion. I checked the primer sequence according to your instruction and correction highlight in color. And the primer for bta-miR-148a was synthesis from the Genewiz. The sequence of RT primer of miRNA is long which also shows in this article (MiR-485 targets the DTX4 gene to regulate milk fat synthesis in bovine mammary epithelial cells).
In "Results", at the point 3.1, instead of an estimate of the transfection frequency at 24 h, the authors show figures of cells expressing gfp, where we can see that the transfection rate is high, but could be any percentage from 50-100. The transfection frequency helps to interpret the differences observed in the following assays, so I consider it important to know it. If the transfection rate is 50%, the observed differences could be corrected by a factor of two.
Ans: Thank you so much for your suggestion. The transfection efficiency in these group between 80-90%. And changing highlight in color in the text.
The 4A, 4B and 13A figures should be improved.
Ans: Thank you for your suggestion. According your suggestion now we revised the figures 4A, 4B and 13A.
The names of the categories in the figures of relative expressions should be simplified. For example, in stead of bta-miR-148a-mimic, bta-miR-148a-inhibitor and bta-miR-148a-shNC , use only miR-148a, anti miR-148a and shNC. The names are too long, and only a few letters are informative.
Ans: Thank you for your suggestion. We wrote complete name rather than abbreviations and short name because we want to give complete information and understanding for the new researchers and avoid the confusion.
The figure 3 is a relative expression. Which expression correspond to expression 1? I don't understand the scale.
Ans: Thank you for your suggestion. In our all research work we used the shNC group as the control group we used the average value rather than the individual value to present the more accurate result. With the average value of the control group the result shows as average value.
Figure 6A and 6B are presented as relative content but with units (umol/ug). Contradictory
Ans: Thank you for your valuable suggestion. Now we revised our mistake and change the figures according to your suggestion.
I suggest huge correction before published

Round 2
Reviewer 1 Report
Dear authors,
Thank you for your revised version.
You are right there was no HDF in your article but HFD. I reversed the letters.
In addition, the structure of your manuscript is still not easy to follow but it is understandable.
Some intermediate conclusions of the sub-paragraphs are a little too affirmative. For example, line 519, I propose to replace identified but "strongly suggest" and the same line 568 replace "demonstrated" by "strongly suggest". Other concluding sentences could be changed in this way.
Author Response
Comments and Suggestions for Authors
Dear Reviewer,
Thank you for your suggestions.
Some intermediate conclusions of the sub-paragraphs are a little too affirmative. For example, line 519, I propose to replace identified but "strongly suggest" and the same line 568 replace "demonstrated" by "strongly suggest". Other concluding sentences could be changed in this way.
Response: Thank you for such a great suggestion. We have revised "demonstrated" into "strongly suggest".

Reviewer 2 Report
My comments regarding Miqe guidelines were not considered by the authors. Some consideration regarding their application in gene expression studies should be added in the comments.
Please carefully check and revise where needed the added phrases regarding fold changes values (some punctuation errors are still present).
Author Response
Comments and Suggestions for Authors
My comments regarding Miqe guidelines were not considered by the authors. Some consideration regarding their application in gene expression studies should be added in the comments.
Response: Thank you for your suggestion. According to our understanding we provided the complete steps for qPCR. During experiment we fallow all the standard protocol( C4BPA: A Novel Co-Regulator of Immunity and Fat Metabolism in the Bovine Mammary Epithelial Cells)( Transcriptome analysis of CRISPR/Cas9-mediated GPAM− /− in bovine mammary epithelial cell-line unravelled the effects of GPAM gene on lipid metabolism).
Primer designing
The sequence of bta- miR-148a was surfed from (http://www.mirbase.org). This study designed primers by Primer Premier 6.0. Each experimental step/reaction was repeated three times, and U6 as a reference. The qPCR primer is shown in (Table ).
Table Real-time PCR primer sequences
|
Name |
Primer Sequence (5'-3') |
|
Bta-miR148a
U6
|
F TGCGGTTCAGTGCACTACAGAA R GTCGTATCCAGTGCAGGGTCCGAGGTGCACTGGATACGACACAAGTT F CTCGCTTCGGCAGCACA R AACGCTTCACGAATTTGCGT |
RNA extraction
The RNA was extracted from cells after 24 hours of transfection through the following protocol:
- Wash the cells with PBS 2 times and then add 1ml AG-RNAase pro-reagent and pipetting 3 -5 times so that cells were sufficiently lysed. Leave it at room temperature for ten minutes. And then collect the sample in their specific 2ml centrifuge tube.
- Add 200 µl of chloroform to the lysate, vigorously shake by hand for 1.5 s, and put at room temperature for 10 min. And then centrifuge at 12000×G for 15 min at 4°C.
- The supernatant was shifted to a new centrifuge tube, added 500 µl of absolute isopropyl alcohol, and vortexed gently. And put at room temperature for 10 min; after that, centrifuge at 12000×G for 15 min at 4°C.
- After centrifugation, throw the whole liquid. And that adds the 500 µl of 80% alcohol and centrifuge at 7500×G for 5 min.
- After that, remove the liquid and put it for 3-5 minutes at room temperature with open covers; then, add 20 µl ddH20 and mix with the pipette.
- Store the collected RNA at -80℃ after checking quality and quantity using nanodrop. Which shows in (Table 2.4).
Table OD and concentration value of the extracted RNA
|
Name |
OD |
Concentration |
|
Bta-miR-148a-mimic |
1.87 |
442.8 |
|
Bta-miR-148a-inhibitor |
1.84 |
362.1 |
|
Bta-miR-148a-shNC |
1.84 |
409.4 |
Reverse transcription reaction
After that, the RNA extraction of the cDNA was performed by the Reverse Transcription Kit. We took the RNA and the reaction mixture made in the table below 1.4. The apparatus used for this is Bio-Rad T100 Thermal cycler PCR. The reaction mixture was incubated for 2 minutes at 42 degrees centigrade and inverse at 25 degrees centigrade for 5 minutes, 50 degrees centigrade for 15 minutes, and the inactivation was done at 85 degrees centigrade for 5 seconds and preserved at -20 degrees centigrade. Which shows in (Table).
Table Reverse transcription reaction mixture
|
Reagents |
Amount |
|
RNA |
1 µg (3/concentration×1000) |
|
5xgDNA wiper mix |
2 µl |
|
RNase -free Water |
to 10 µl |
Put for 2 minutes at 42 degrees centigrade
|
10RT mixture |
2 µl |
|
HiScript II enzyme mixture |
2 µl |
|
148 RT Primer/U6 Primer/ Syber green mixture |
1 µl |
|
RNase -free Water |
5 µl |
|
Total |
20 µl |
qPCR
The kit was used for the qPCR (Vazyme, CUSqMM, Q711, China). The reaction mixture was prepared in the 96 well plates by the given protocol in table 1.6.
The real-time quantitative qPCR consists of a two steps experiment. The pre-degeneration process took place at 95 degrees Celsius for 5 minutes. At 95 degrees Celsius for 10 seconds, then at 60 degrees Celsius for 30 seconds, extend the degeneration process for another 30 seconds, totaling 42 cycles. Each cycle rose by 0.5 degrees Celsius. Finally, the temperature rose to 95 degrees Celsius for 15 seconds and was read on the final plate reading.
Real quantitative results are calculated and data analysis of the final values in Microsoft Excel through 2-ΔΔ CT using the GAPDH as a reference gene. Which shows in (Table 2.6).
Table shows the qPCR reaction mixture
|
Reagents |
Amount |
|
2×miRNA Universal SYBR qPCR Master Mix |
10 µl |
|
Forward Primer |
0.4 µl |
|
Reversed Primer |
0.4 µl |
|
cDNA |
1 µl |
|
ddH2o |
to 20 µl |
qPCR
The primer sequence shows in (Table ).
Table Primer for Real-time PCR
|
Name |
Primer Sequence (5'-3') |
|
KLF6
GAPDH |
F CTCGAGTCTAGCTGTTAATGCACTGTAGATCTTAATAAT R GGCCGCATTATTAAGATCTACAGTGCATTAACAGCTAGA F GCATCGTGGAGGGACTTATGA R GGGCCATCCACAGTCTTCTG |
Dual-luciferase activity
Cow (ARS-UCD1.2) â–¼
Transcript: ENSBTAT00000081800.1 KLF6-202
Description Location
Kruppel like factor 6 [Source:VGNC Symbol;Acc:VGNC:56191 ] P rimary_assembly 13: 44,597,240-44,605,699 forward strand.
About this transcript
This transcript has 4
exons, is annotated with 1 7 domains and features, is associated
Gene
with 1 220 variant alleles and maps to 5 1 oligo probes.
This transcript is a product of gene E NSBTAG00000015188.5 Hide transcript table
|
Show/hide columns (1 hidden) Filter |
||||||
|
Transcript ID |
Name |
bp |
Protein |
Biotype |
UniProt Match |
Flags |
|
E NSBTAT00000020207.5 |
KLF6-203 |
3842 |
2 67aa |
P rotein coding |
J 9JH87 |
APPRIS P1 |
|
E NSBTAT00000081800.1 |
KLF6-202 |
4310 |
2 80aa |
P rotein coding |
A 0A3Q1M9X5 |
- |
|
E NSBTAT00000087087.1 |
KLF6-201 |
5639 |
3 09aa |
P rotein coding |
A 0A3Q1M439 |
Ensembl Canonical |
cDNA sequence
Download sequence BLAST this sequence
|
Codons Alternating codons Alternating codons
Exons An exon Another exon
Variants 3 prime UTR 5 prime UTR Missense Synonymous
Other
Markup loaded
Variants are filtered by consequence type |
|
UTR |
Y R Y M K K
1 CGGGGAGGAGGGGAGGCTGGCAGCGGAGCTTTGAATAGGGAAGTTTTGCAGGGGTTACGT 60
............................................................
............................................................
K K K K
W S
M M S
S Y
61 T TGCAGTCAGTCCGGTGTTTGCAAATATTGTGCGGGCTCGCGAGCGCGTCTCCGGGCTCC 120
............................................................
............................................................
S S S K
M M
Y R K
M M W
121 G GCCAGGACCCGAACCCGCGGCGCCTAATCGCTGCGCACTTGAGTTTGCATGAACTTCCC 180
............................................................
............................................................
S K
K YKS
V K MS
S V Y R
M S
M KM S
M Y *
181 C GGCGCTGCAGGCACGGCCGCCGCGCTCCCGACTGCAGACCGCAAGCCTCCCTGTTTTTA 240
............................................................
............................................................
M S D W
R K M
M K R M
241 CAGCAGCGGGGACGGTCTCTTCCAACCCGACATGGATGTGCTCCCAATGTGCAGCATCTT 300
............................................................
............................................................
S Y M
W DYS
M MR
R MK K
K S K
301 CCAGGAACTCCAGATTGTGCACGAGACTGGTTACTTCTCGGCGCTGCCCTCCCTGGAGGA 360
............................................................
............................................................
M
361 A TACTGGCAACAGTGGCTAGAAAGTTTGAGATGGTGGGGAGAGGGGATGCTGGTGTCTCG 420
..............................ATGGTGGGGAGAGGGGATGCTGGTGTCTCG 30
..............................-M--V--G--R--G--D--A--G--V--S- 10
S Y S
421 GCAGTTGTAGCTTCTGGAAAGACCTGCCTGGAGTTGGAACGTTACCTGCAGAGCGAGCCC 480
31 GCAGTTGTAGCTTCTGGAAAGACCTGCCTGGAGTTGGAACGTTACCTGCAGAGCGAGCCC 90
11 -A--V--V--A--S--G--K--T--C--L--E--L--E--R--Y--L--Q--S--E--P- 30
R S M Y M
481 TGCTACGTGTCAGCCTCCGAGATCAAATTCGACAGCCAGGAAGATCTGTGGACCAAAATC 540
91 TGCTACGTGTCAGCCTCCGAGATCAAATTCGACAGCCAGGAAGATCTGTGGACCAAAATC 150
31 -C--Y--V--S--A--S--E--I--K--F--D--S--Q--E--D--L--W--T--K--I- 50
K K
Y Y K
S S K M
Y Y S
541 ATCTTGGCTCGGGAGAAAAAGGAGGACGCGGACCTGAAGGTGTCCTCCTCTGGCCCGCCC 600
151 ATCTTGGCTCGGGAGAAAAAGGAGGACGCGGACCTGAAGGTGTCCTCCTCTGGCCCGCCC 210
51 -I--L--A--R--E--K--K--E--D--A--D--L--K--V--S--S--S--G--P--P- 70
S RR
R MK Y
M M
S M M
601 G AGGACGCGCTCCACAGCCCTGGCTTCAGCTACAACCTGGAGACCAACAGCCTGAACTCG 660
211 GAGGACGCGCTCCACAGCCCTGGCTTCAGCTACAACCTGGAGACCAACAGCCTGAACTCG 270
71 -E--D--A--L--H--S--P--G--F--S--Y--N--L--E--T--N--S--L--N--S- 90
Y M
Y W
M SY
S M
661 GATGTGAGCAGCGAGTCCTCCGACAGCTCCGAGGAGCTCTCGCCCACCACCAAGTTTACC 720
271 GATGTGAGCAGCGAGTCCTCCGACAGCTCCGAGGAGCTCTCGCCCACCACCAAGTTTACC 330
91 -D--V--S--S--E--S--S--D--S--S--E--E--L--S--P--T--T--K--F--T- 110
M M Y
V S
K M
K R M R
M W
721 TCCGACCCCATCAGCGAAGTCTTGGTCAATTCGGGGAATCTGAGCTCCTCGGTCACCTCC 780
331 TCCGACCCCATCAGCGAAGTCTTGGTCAATTCGGGGAATCTGAGCTCCTCGGTCACCTCC 390
111 -S--D--P--I--S--E--V--L--V--N--S--G--N--L--S--S--S--V--T--S- 130
H R K
K M
S S
M M K K
K K M
|
781 A CGCCTCCGTCCTCCCCAGAACTGAGCAGGGAGCCCTCGCACCTGTGGGGCTGTGTGCCT |
840 |
|||
|
391 ACGCCTCCGTCCTCCCCAGAACTGAGCAGGGAGCCCTCGCACCTGTGGGGCTGTGTGCCT |
450 |
|||
|
131 -T--P--P--S--S--P--E--L--S--R--E--P--S--H--L--W--G--C--V--P- |
150 |
|||
|
N |
S S |
R S |
K M |
|
|
841 GGCGAGCTGCACCCCCCCGGGAAGGCGCGGGGCGGGACCTCAGGGAAGCCGGGCGACAAG |
900 |
|||
|
451 GGCGAGCTGCACCCCCCCGGGAAGGCGCGGGGCGGGACCTCAGGGAAGCCGGGCGACAAG |
510 |
|||
|
151 -G--E--L--H--P--P--G--K--A--R--G--G--T--S--G--K--P--G--D--K- |
170 |
|||
M R MR W S K S
|
901 |
GGCAGCGGGGACGCCTCCCCCGACGGCCGCAGGCGGGTGCATCGCTGCCACTTTAATGGC |
960 |
|
511 |
GGCAGCGGGGACGCCTCCCCCGACGGCCGCAGGCGGGTGCATCGCTGCCACTTTAATGGC |
570 |
|
171 |
-G--S--G--D--A--S--P--D--G--R--R--R--V--H--R--C--H--F--N--G- |
190 |
|
961 |
TGCAGGAAAGTTTACACCAAAAGCTCCCACTTGAAAGCACACCAGCGCACCCACACAGGA |
1020 |
|
571 |
TGCAGGAAAGTTTACACCAAAAGCTCCCACTTGAAAGCACACCAGCGCACCCACACAGGA |
630 |
|
191 |
-C--R--K--V--Y--T--K--S--S--H--L--K--A--H--Q--R--T--H--T--G- |
210 |
|
1021 |
GAAAAGCCTTACAGATGCTCATGGGAAGGGTGTGAGTGGCGTTTTGCAAGAAGCGATGAG |
1080 |
|
631 |
GAAAAGCCTTACAGATGCTCATGGGAAGGGTGTGAGTGGCGTTTTGCAAGAAGCGATGAG |
690 |
|
211 |
-E--K--P--Y--R--C--S--W--E--G--C--E--W--R--F--A--R--S--D--E- |
230 |
|
1081 |
M Y TTAACCAGACACTTCAGAAAGCACACTGGTGCCAAGCCTTTTAAATGTTCTCACTGTGAC |
1140 |
|
691 |
TTAACCAGACACTTCAGAAAGCACACTGGTGCCAAGCCTTTTAAATGTTCTCACTGTGAC |
750 |
|
231 |
-L--T--R--H--F--R--K--H--T--G--A--K--P--F--K--C--S--H--C--D- |
250 |
|
1141 |
K R AGGTACGTGCATGAACCACAGGGGCGGGGGTGCCATCCCCGGGAGCAGGCATGCCGTCAC |
1200 |
|
751 |
AGGTACGTGCATGAACCACAGGGGCGGGGGTGCCATCCCCGGGAGCAGGCATGCCGTCAC |
810 |
|
251 |
-R--Y--V--H--E--P--Q--G--R--G--C--H--P--R--E--Q--A--C--R--H- |
270 |
|
1201 |
K Y K W CCCGCAGCGGGCAGCCATCCTCCGGGCCTGTAGTGGCATTTCTCTTTTCAGCCCAGGACT |
1260 |
|
811 |
CCCGCAGCGGGCAGCCATCCTCCGGGCCTGTAG........................... |
843 |
|
271 |
-P--A--A--G--S--H--P--P--G--L--*-........................... |
280 |
W B M S S Y
1261 TTTTTAAAAAGGCTCTTGTTTGCCAGTCCTCCGGGTTAGGAAGCCCCAGGTACTTGAGTG 1320
............................................................
............................................................
Y K K
1321 CCTGAGAATAGGGTGATTTCGTTCATACTTTCCGAGGAACAACCCTTAATCAGTTTTCAG 1380
............................................................
............................................................
K
1381 ATTTGAAGGCCAGGTAGTAAAGTCTAGTGACATTCCCAAGAACTCACTGGTGGCCTCTTT 1440
............................................................
............................................................
R S R K K M S
1441 GATGTAGATACTTCTGTCAGTTGGAATTTCCTGCCCCAAATGAGGATGGGGTGACTCAGC 1500
............................................................
............................................................
K M
K K
S R S
R B
R R M
R R S K
1501 TGGGGGAGTGCACCCTCCTGGGGAGAGGCTGAGGTTGGGGAGCTCGGGGCAGAGGTCTGT 1560
............................................................
............................................................
W M R
K M Y
W K
W Y
K K M
1561 GACCTGCAGGGCAGGAGAACTGATGTCTGGGACCCCGATTCTCTCCACAGGTGTTTCTCC 1620
............................................................
............................................................
Y S
W S
K S
R KSY K M
1621 AGGTCCGACCACCTGGCCCTGCACATGAAGAGACACCTCTGAGGGGGCAGAGCGACAACT 1680
............................................................
............................................................
R M
S M
R B K
1681 CCTGCGGGCTAAAAAGGCTTCCAGGCTGAGACGGCCGCGGAAGGAGGGTGCGTGTTCCAG 1740
............................................................
............................................................
M Y
1741 CCAAAGCCATTTTGCACCCTGCCCGGTCACCTCCAGGACCTACTGGGAAGGTCTTTCGAG 1800
............................................................
............................................................
R Y
K M M
K K K
K Y
K R
R K
1801 GGCGAGAACAGTCACGTCCGAAGCGGCAGGGCACCCGGGGTGTGTGGTGTCTGGGTGGCC 1860
............................................................
............................................................
Y Y
M K
R K V
1861 CTGGCTCGCCACTGGTTACCTGTCCTCTGAGTGGTCCCTAAGCCTTTGCCGTGAGCATGT 1920
............................................................
............................................................
K K K
K K
* *MYYSB M
K H
1921 GCACTGAGAATGTTAGTGGTGGGTGGGCTGTATGTGGAGGGTCTCCTACGGACTGGGTAA 1980
............................................................
............................................................
M S R M S K
1981 CGAGGCACATCCTTTCCCTGGCGAGACTGCAAGAGACAGAAAAAAAAAAACAGAGTTTGA 2040
............................................................
............................................................
K K K Y
2041 ATGTTCTGTGTGTGAGGCGTGTTCCGTGAGACGAGATGACCATCCTTCATTTCCTGAGGG 2100
............................................................
............................................................
K *
M R
K K
2101 GGTGTCTGCGCCTTAGAGGAGAGAAAAAAAAAAAGACAAGAAGAAGGGAAGATGTGGAGG 2160
............................................................
............................................................
K K B K
|
2161 |
GCAGGCGTGGGCAGTCAGGACCCCGAGTGACGAGTGGGTGTTGGTGGGGAGAGCCCCTTC ............................................................ ............................................................ |
2220 |
|
2221 |
CCCCGCGGTGTGTGACGGGAGCGTTCGGTGTTCGGTTCAGCTGTGTGAAGTGAGTAGCTG ............................................................ ............................................................ |
2280 |
|
2281 |
R R GCGCTGCTTGCTGTGACGGAGTTCATTTGTTACCTGTTGTATACGCTGTGTACCTACAAA |
2340 |
|
|
............................................................ ............................................................ |
|
M
2341 CATAACACGTGGATAACTTTTCCTTCTAAGACAAAAGTAATGCATGGTCTGGGGAATTAT 2400
............................................................
............................................................
Y
2401 CTGCTTTTCCATTTTTAAATCAGGACTACCGTTTTGGTTCCAAGTTCACTGAGCAGCTAA 2460
............................................................
............................................................
Y M Y
2461 GATAACAATTGGTCGAATCAGGTGACCCCCTAGCCATCTGGGCCTGGCAATACATGATGC 2520
............................................................
............................................................
M M Y K R
2521 TCCACCCCCCCCAACTCCCGCCCCTTGGATTTATTCGTAACACTCCCCTTCTAGAGAGGC 2580
............................................................
............................................................
V K YK Y
2581 GTTGTGCAGTGTAGGACTATTTTGTCAGTGGTTCCGAATTCGTATTACCTTTTTGGAGAA 2640
............................................................
............................................................
Y Y
2641 TTTTAAGTTGCCCTTGGGAGACATTGGGCACGAGTTGGGTGTCCGGGACACCAAGGCTGG 2700
............................................................
............................................................
Y Y
2701 GAATCGCTGGTCTAACGTCCCACTAGACTCAGAACCTAAAATTTTTCTCAGTTGGGTGGA 2760
............................................................
............................................................
M
2761 TGAAACCACTAAAGCTTAGAAACTGTTTTCTCATGCAGCTAAGTTTCTCTTATTTATGCC 2820
............................................................
............................................................
K R
2821 TTGAGGACTCATTTCTGGTTTTCTAGCTGTTAATGCACTGTAGATCTTAATAATGGTGCC 2880
............................................................
............................................................
R S
K M
R K
S K W K W Y D
2881 TTACGCAAGTGGCCCCTCCTGTGGGGGTCTCATACAGGGGAATGGGTGATGCATGCTTTA 2940
............................................................
............................................................
K Y Y
Y M
Y K
M Y
W W M
* ** W
2941 TTAAGGCTCTTTTTTCACCTGGCTTTGTACTGTATCAATATTTAATACAGAAAAAAAAAA 3000
............................................................
............................................................
M M MMM
Y Y
Y Y W Y
* M
R B
W R R D Y
3001 AAAACAAAACTCTTTAAGGTCCTACTTCACAAGTCAAATAGATGAAAACTCCCCAAACAA 3060
............................................................
............................................................
K R Y
3061 GTCAGTATTTACTCAATAGCTTAGACAACTTGACTGTCAGTGTTCAAGTGGGAAACACAT 3120
............................................................
............................................................
R M
3121
G ATAATTGGAAAATCAGACCATTTCTGCCACCGTGAGGGTGAGACTTCAATATAGACTTT 3180
............................................................
............................................................
S *
3181 GCACAACACTTGTACAGATCATACACCGGCTGTGTTTAATATGTAACATTTTCACACGTA 3240
............................................................
............................................................
Y S
3241 TTAAAGATACAGAAGTATTAAAAAACAATGTATTTGCTTAAAAGGCACAAGTTTCTCGCG 3300
............................................................
............................................................
M Y
3301 TATCTATCTAGCTATCTGTTGGTAACACAGAAGGTATATTACTTTTTTAAAAAGTGGGCG 3360
............................................................
............................................................
* ** K
Y R K V
W K *
3361
A AATTCTTGTGTATGTACATTTGTGTGTATAATATGTGTATGTGTGTGTGTATATATATA 3420
............................................................
............................................................
Y Y M
K K SM
W H
3421 TATACATATATATGAATAATATATAAATATTTTTTTAAGGAGAAATTAGAATGTATAGCT 3480
............................................................
............................................................
M K
R H
R W
R R W
M W R
3481 AGAAAATTCCACAGCCTGTGAAGACCTATTTCAAAATGGCCATAAAGGAGGTAAAAATGA 3540
............................................................
............................................................
Y D YYKK S K M
3541 AAACTATAACTTTTATAAAGGCTTTATCTTTCATTTAACAATTTGCCACAGGCCTTTGGC 3600
............................................................
............................................................
R S M K
3601
A TCAGATCTCTTCCAGACTGCCAAGTCACTCATCATAATAGGTGGGTCTGGGATAGGGTC 3660
............................................................
............................................................
K M K K
3661 CCTTGGCCATTCAGAACTGCCATGAACGGAATACTAGCCCGCTGGTTGTTTGGTCTCAGA 3720
............................................................
............................................................
W K D R Y W SW W
3721 CACACTAAGGTTTTGATTTCAAATGTCAGCCTTATTGAAGATCTAACCTAAAAGAAAGCT 3780
............................................................
............................................................
Y R
Y M
K Y
K Y S
3781 AATCACAGAGATTCTTTTTGAGAGCATTTTTTATGCAGATGAAGCTACTTTTTTTCCAGC 3840
............................................................
............................................................
M R
W K
Y R W K
3841 ACATAGATTCTTCCAGTTTTTCCAACAAGTAATTTCCCCGAATTGGCATACTGCAGCATG 3900
............................................................
............................................................
R Y
M K
S Y R
3901 GACAGCTGATCCTCCATGCAGCTGCTGGTCAAGGGTGTGGATCTTCTCTTTATATATATT 3960
............................................................
............................................................
* ** R
* *
* *
S K
K R
3961 TATTTATATATATATATATACATATATATATACACATAAGTCTGGCTGGGCTGGTATTTT 4020
............................................................
............................................................
K * *
M M
R * ***********
4021
G TTTGATCTTCCTGAAAATGAGCAATGGTAAAAGCTCATATAACTGGTTTTTTTTTTTTT 4080
............................................................
............................................................
W WWWW
W DW**
* KKK**KBDK
W M MH
B H**H
R M Y W
B Y
D R *
4081
T TTTTTTTAAATCTGGGCTGATGGATACACTTACGTAAAGAAATTATTTTGCTGAGGGGG 4140
............................................................
............................................................
* W R
S MYY W
Y K
K S W
Y Y
K W
4141
A AAATGCACTTTTTTCTTTTGGCAGTTCCAAATCCAGACACTTGATTTGCTGGATTTTGG 4200
............................................................
............................................................
S S R
K Y
K B
K S M Y
4201
G GGAAAAAAACTCCTTTTTTGGCCCTGCTGTGTGCTTAGCCATAACAGTTCCATTAAGCA 4260
............................................................
............................................................
R K W
* *
* *
B
4261 AGAAGGTAAACAAAAGACAAAAAATAAAAAACTTGCACTGGCTTGTCTCA 4310
..................................................
................................................
Primer designing
The binding site of the target gene KLF6 at bta-miR-148a through predicated through Target Scan (http://www.targetscan.org). And then used the ensemble software for designing the KLF6-Wild-Type and KLF6-MUTANT primers. Primer sequences for dual-luciferase reporter assay shows in (Table ).
Table Primer sequences for dual-luciferase reporter assay
|
Name |
Primer Sequence (5'-3') |
|
KLF6-WT
KLF6-MUT |
F GGCCTCTTCCAGGAACTCCAGATTGTGCACGAGACTGGTTACTTCTCGGCGCTGCCCTC R TCGAGAGGGCAGCGCCGAGAAGTAACCAGTCTCGTGCACAATCTGGAGTTCCTGGAAGA F GGCCTCTTCCAGGAACTCCAGATTGTGCGATCGACTGGTTACTTCTCGGCGCTGCCCTC R TCGAGAGGGCAGCGCCGAGAAGTAACCAGTCGATCGCACAATCTGGAGTTCCTGGAAGA |
Bta-mIR-148a target the KLF6 mRNA by specifically binding to its 3’-UTR
Statistics and analysis
The means and standard error of the mean (SEM) of separate experiments were used to present all results (n ≥ 3). The significant differential analysis among different groups was performed using unpaired t-tests in GraphPad Prism8 software (San Diego, CA, USA). Statistical significance is presented as ∗p <0.05, ∗∗p <0.01, ∗∗∗p <0.001.
Please carefully check and revise where needed the added phrases regarding fold changes values (some punctuation errors are still present).
Response: Thank you for highlighting the deficiency. We have revised the language in the whole manuscript.

Reviewer 3 Report
If the authors want to keep all the results presented in this paper, the title and the introduction must be rewritten, putting something similar to: KLF6 is a pivotal gene in the regulation of milk fat that could be regulated by Bta-miR-148a.
Figures 4A, 4B and 13A have not good quality.
Author Response
Comments and Suggestions for Authors
If the authors want to keep all the results presented in this paper, the title and the introduction must be rewritten, putting something similar to: KLF6 is a pivotal gene in the regulation of milk fat that could be regulated by Bta-miR-148a.
Response: Thank you for such a great suggestion. We have revised the title and introduction based on your suggestion.
Figures 4A, 4B and 13A have not good quality
Response: We have provided the increased quality images for these figures.
